# Concept Removal for Frontier Image Generative Models

**Aditya Kumar** [1]    **Pierre Joly** [1]    **Adam Dziedzic** [1]    **Franziska Boenisch** [1]

## Abstract

Image generative models are trained on massive, largely uncurated internet-scale datasets that contain undesirable visual concepts. Efficiently removing such concepts from the model generations without degrading the quality of output images remains challenging. We introduce a novel concept removal method for frontier diffusion and image autoregressive models, such as, SD3.5, Flux, and Infinity. Our intervention replaces the internal bottleneck layer present in all these modern models with a transcoder that is trained to replicate the original layer while structuring it into distinct activation features. This in-place substitution creates an integrated filter through which concept-specific signals can be selectively disabled while preserving the rest of the model's behavior. Since the intervention modifies the model backbone rather than attaching an external component, it remains persistent under white-box access. Empirically, the approach achieves state-of-the-art concept removal performance across modern diffusion and autoregressive models, maintains visual generation quality, provides robustness against adversarial prompts, and supports sequential removal of diverse concepts. This positions our method as a practical approach for concept removal in frontier image generative models[1].

## 1. Introduction

Image generative models, such as diffusion models (DMs) (Rombach et al., 2022; Esser et al., 2024b; StabilityAI, 2023; Black Forest Labs, 2024) and image autoregressive models (IARs) (Tang et al., 2024; Han et al., 2025), have revolutionized the creation of visual content. The latest frontier models, such as, SD3.5 (Esser et al.,

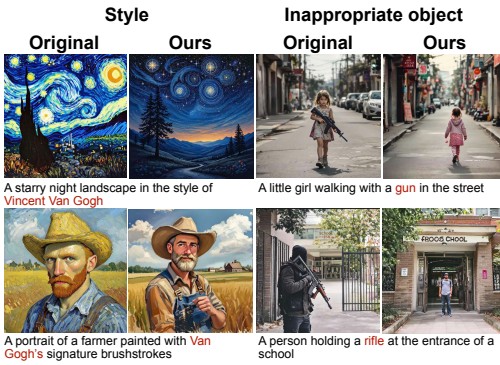

**Style**
Original    Ours

**Inappropriate object**
Original    Ours

A starry night landscape in the style of Vincent Van Gogh

A little girl walking with a gun in the street

A portrait of a farmer painted with Van Gogh's signature brushstrokes

A person holding a rifle at the entrance of a school

*Figure 1.* **BLOCK removes undesirable artistic styles and inappropriate objects from generated images.**

2024b), FLUX (Black Forest Labs, 2024), and Infinity (Han et al., 2025) generate photorealistic, extremely detailed, and highly-aesthetic content. Despite their capabilities, these models often raise concerns, as they can inadvertently generate undesirable content (Gary Marcus, 2024; Qu et al., 2023; Rando et al., 2022; Yang et al., 2024b), including copyrighted artistic styles (Asperti et al., 2025) or inappropriate objects (Wei et al., 2025).

There exist two types of approaches for removing such unwanted concepts in image generative models, namely internal and external ones. The former *alters the model in-place*, either through training or through direct closed-form model edits. Training-based approaches (Zhang et al., 2025; Gandikota et al., 2023; Gao et al., 2025; Zhang et al., 2024a; Kumari et al., 2023; Wu et al., 2025; Fan et al., 2024; Wu & Harandi, 2024; Heng & Soh, 2023; Lu et al., 2024) suffer from high computational costs, making them unsuitable for the frontier models. Additionally, they usually rely on objective functions tied to a specific training paradigm, which restricts their applicability across models. Closed-form edits (Orgad et al., 2023; Gandikota et al., 2024; Basu et al., 2024a; Gong et al., 2024) are computationally efficient and can be applied across models but tend to degrade generation quality, especially under sequential concept removal requests, which is limiting in real-word deployment where multiple undesired concepts can be identified over time.

The latter external type of approaches *adds concept-removal* modules either at inference time (Schramowski et al., 2023; Li et al., 2024) or by attaching *additional* learned components to the model. In particular, sparse autoencoder

---

[1]CISPA Helmholtz Center for Information Security, Germany. Correspondence to: Aditya Kumar <aditya.kumar@cispa.de>.

*Proceedings of the 43rd International Conference on Machine Learning*, Seoul, South Korea. PMLR 306, 2026. Copyright 2026 by the author(s).

[1]https://github.com/sprintml/block-concept-removal

(SAE)-based methods (Cywiński & Deja, 2025) suppress internal features for concept removal. However, they remain *external* to the generative backbone and can be detached or bypassed under white-box access, limiting their suitability for open-source frontier models.

To address these shortcomings, we introduce **B**ottleneck-**L**ayer-**O**riented-**C**oncept-**K**nockout (BLOCK), a new framework for concept removal in frontier image generative models. Our key observation is that these most-advanced diffusion and autoregressive models contain a narrow transformation layer between the text encoder and the generative backbone. This layer serves as an architecturally-determined bottleneck through which all conditioning signals must pass. BLOCK replaces this bottleneck layer with a transcoder (Dunefsky et al., 2024), a lightweight module trained to replicate the original layer while exposing structured activation features. This *integrated* substitution provides a *single, unavoidable intervention point* at which concept-associated features can be identified and selectively disabled before they propagate into the model. Importantly, unlike add-on safety modules, including SAE-based ones (Poppi et al., 2024; Cywiński & Deja, 2025; Schramowski et al., 2023; Li et al., 2024), that can be detached or bypassed, our method integrates the transcoder directly into the generative model's backbone where it remains persistent even under white-box access. Most importantly, our method eliminates the need for backpropagation through the full model as the transcoder can be trained in isolation at low computational costs. This enables our method to scale to frontier diffusion and autoregressive architectures with billions of parameters.

We evaluate our method on state-of-the-art (SOTA) DMs, namely Stable Diffusion 3.5 Large (SD3.5) (Esser et al., 2024a) and FLUX.1-dev (Flux) (Labs, 2024), as well as on IARs, namely Infinity-2B and Infinity-8B (Han et al., 2025). We focus on **style** and **object removal**, following the UnlearnCanvas benchmark (Zhang et al., 2024b). Our results highlight that BLOCK achieves SOTA performance in concept removal across both paradigms, DMs and IARs. For example, on our largest model Flux (12B parameters) it surpasses the baselines by up to 21 percentage points in style removal accuracy, while also achieving significantly higher in- and cross-domain retain accuracy. This trend is consistent over all tested models, confirming our method's flexibility and scalability. In Figure 1, we also provide qualitative results for Infinity-8B which demonstrate that our method effectively removes diverse concepts while maintaining high visual quality. We further show that unlike prior weight-editing methods that degrade under sequential edits, BLOCK remains effective when removing concepts sequentially. Finally, BLOCK maintains robust concept removal also under adversarial prompting strategies. Together, these results establish BLOCK as a practical and general solution for concept removal in modern image generative models. In summary, we make the following contributions:

- We introduce BLOCK, a method that replaces the bottleneck layer in frontier image generative models with a transcoder, enabling *efficient and persistent* concept removal, even under white-box access.

- We demonstrate BLOCK's broad applicability and its effectiveness for concept removal in both frontier diffusion and autoregressive models.

- We show that BLOCK supports efficient sequential and multi-concept removal, a setting that highly challenges existing closed-form and training-based approaches.

- Our extensive experiments show that BLOCK achieves strong concept removal while maintaining high visual fidelity even under diverse adversarial prompting strategies.

## 2. Related Work

**Text-to-Image Models.** Currently, there are two dominant paradigms for text-to-image modeling, namely Diffusion Models (DM) and Image AutoRegressive (IAR) models. DM learn the gradient field between Gaussian noise and the image distribution, originally through stochastic processes (Rombach et al., 2022), with recent work favoring deterministic formulations (Esser et al., 2024a). IAR models instead factorize the joint distribution into conditional probabilities, generating tokens sequentially. While early IARs were restricted to class-conditional generation (Van Den Oord et al., 2016; Ramesh et al., 2021), recently, models like Infinity (Han et al., 2025) are text-conditional and generate images at SOTA quality based on the principle of next-scale prediction. See Section A.1 for more details on both paradigms. Our proposed BLOCK method applies to both DMs and IARs.

**Concept Removal** aims to prevent a model from generating content related to a *target concept*. Approaches can be grouped into two categories: **internal** methods, which directly modify model weights or structure, and **external** methods, which leave the model frozen and operate externally, for example by adding separate, removable modules (Cywiński & Deja, 2025) or modifying generation at inference time (Schramowski et al., 2023; Li et al., 2024). We present a full taxonomy in Table 12 and an extended descriptions of the relevant methods in Section A.2.2.

**Internal** concept-removal methods modify the model's parameters directly and can be divided into two major families: training-based editing and closed-form editing. **Training-based methods** (Zhang et al., 2025; Gandikota et al., 2023; Gao et al., 2025; Zhang et al., 2024a; Kumari

et al., 2023; Wu et al., 2025; Fan et al., 2024; Wu & Harandi, 2024; Heng & Soh, 2023; Lu et al., 2024) fine-tune parts of the model to suppress a target concept. While often effective, they require substantial computational resources, depend on curated training data, and are typically tied to architecture-specific objectives. In particular, most evaluations focus on U-Net-based DMs, limiting applicability to modern large-scale transformer-based DMs and IARs. **Closed-form methods** (Orgad et al., 2023; Gandikota et al., 2024; Basu et al., 2024a; Gong et al., 2024) instead apply analytic projections to cross-attention parameters of the models without retraining. These techniques are efficient but exhibit well-known trade-offs: weak edits leave residual traces of the concept, whereas strong edits degrade image fidelity. Moreover, because they modify weights throughout the network without accounting for polysemantic neurons (Elhage et al., 2022), they struggle under sequential or multi-concept edits, where repeated projections accumulate and distort model behavior. BLOCK differs from both families by avoiding broad weight updates entirely: rather than training or projecting across the model, it replaces a single, architecture-mandated text-to-backbone transformation layer with a transcoder. This yields precise, persistent edits while maintaining the computational efficiency of closed-form approaches.

**External** concept-removal methods do not modify the underlying model and therefore remain *non-persistent under white-box access*. Some approaches intervene only during **inference** (Schramowski et al., 2023; Li et al., 2024), altering the denoising trajectory or text conditioning on a per-prompt basis. These methods are flexible but can be trivially bypassed by removing the additional inference-time logic. Another approach, namely SAeUron (Cywiński & Deja, 2025), attaches **sparse-autoencoder** (SAE)-based modules to the model to sparsify and edit internal representations for feature removal. Yet, SAeUron was designed for U-Net-based DMs, and its applicability to newer transformer-based architectures and IARs remains open. Additionally, while the method provides interpretability and fine-grained control on the features to be removed, it remain external components: the generative backbone itself is untouched, and removing the SAE restores the model's original behavior. Although BLOCK also relies on disentangling features for targeted removal, it differs fundamentally from SAE-based approaches: instead of attaching removable modules, BLOCK permanently replaces a single, *mandatory* text-to-backbone transformation layer. This makes our intervention non-removable even under white-box access.

## 3. Background and Notation

**Transcoders** are neural networks designed to learn a *sparsified* approximation of multi-layer perceptron (MLP) layers.

They promote sparsity by adding an $\ell_1$ penalty on latent activations to the reconstruction loss, yielding a more interpretable latent representation.

Formally, let $\mathbf{x}, \mathbf{y} \in \mathbb{R}^d$ denote an input–output pair of an MLP layer (*i.e.*, $\mathbf{y} = \mathrm{MLP}(\mathbf{x})$). A transcoder maps the input $\mathbf{x}$ into a higher-dimensional latent representation $\mathbf{z} \in \mathbb{R}^{m \cdot d}$, where $m$ is the expansion factor, from which it reconstructs the output $\hat{\mathbf{y}} \in \mathbb{R}^d$. The architecture of the transcoder can be formalized as:

$$\begin{aligned} \mathbf{z} &= \mathrm{ReLU}(W_{\mathrm{enc}}\mathbf{x} + \mathbf{b}_{\mathrm{enc}}), \\ \hat{\mathbf{y}} &= W_{\mathrm{dec}}\mathbf{z} + \mathbf{b}_{\mathrm{dec},} \end{aligned} \tag{1}$$

where $W_{\mathrm{enc}} \in \mathbb{R}^{m \cdot d \times d}$ and $W_{\mathrm{dec}} \in \mathbb{R}^{d \times m \cdot d}$ are encoder and decoder weight matrices, respectively. The $\mathbf{b}_{\mathrm{enc}} \in \mathbb{R}^{m \cdot d}$ and $\mathbf{b}_{\mathrm{dec}} \in \mathbb{R}^{d_2}$ are learnable bias terms. Through this formulation, each feature in the transcoder is associated with two vectors, namely the $i$-th row in $W_{\mathrm{enc}}$ (encoder feature vector) and the $i$-th column of $W_{\mathrm{dec}}$ (decoder feature vector). Intuitively, for every feature, the encoder vector indicates how much the feature should activate and the decoder vector is scaled by this amount. The resulting weighted sum of the decoder vectors represents the transcoder's output. Transcoder training minimizes the reconstruction objective of the original MLP with an $\ell_1$ regularization on the latent:

$$\mathcal{L}_t = \underbrace{\|\hat{\mathbf{y}} - \mathbf{y}\|_2^2}_{\text{faithfulness loss}} + \underbrace{\lambda \|\mathbf{z}\|_1}_{\text{sparsity penalty}}, \tag{2}$$

where $\lambda > 0$ controls the trade-off between faithfulness and sparsity (Dunefsky et al., 2024).

**Notation.** We denote by $t$ a target concept to be removed and by $p$ a prompt used to generate an image with the text-to-image model $\mathcal{M}$. The prompt consists of $P$ tokens $\{p_1, \ldots, p_P\}$, and we denote the tokens that encode concept $t$ by the subset $p_t = \{p_{t1}, \ldots, p_{tk}\}$, where $k$ is the number of tokens for concept $t$. For example, in the prompt $p =$"*An image of a cat in Van Gogh style*", with concept $t =$"*Van Gogh*", $k = 2$ and $p_{t1} =$"*Van*", and $p_{t2} =$"*Gogh*".

## 4. Transcoders-based Concept Removal

In this section, we present BLOCK, the first application of transcoders for concept removal in text-to-image generative models. While transcoders have previously enabled analysis of MLP circuits in large language models (Dunefsky et al., 2024), their use for concept removal, in particular for text-to-image architectures, remains unexplored. We leverage transcoders' ability to sparsely approximate and replace SOTA image-generative models' inherent transformation bottleneck layers to selectively suppress undesired concepts with minimal side effects. We first provide a high-level overview of our framework, then describe transcoder training and our targeted intervention for concept removal.

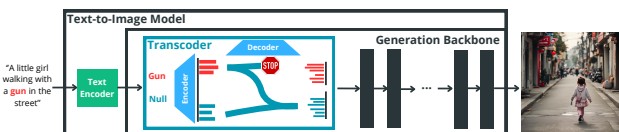

*Figure 2.* **Overview of our BLOCK.** We detail our transcoder-based concept removal framework.

### 4.1. Our BLOCK Framework

SOTA architectures, including both DMs and IARs, rely on one or more text encoders that generate embeddings to guide image generation. These embeddings are typically injected into the image-generative backbone via transformations such as projection layers or MLPs applied to pooled text features, depending on the specific model (see Section D.1 for details). In all cases, we can uniformly replace this transformation with a transcoder that yields condensed, monosemantic features from text representations, enabling precise suppression of targeted content. We present a schematic overview of our approach in Figure 2.

By intervening at this early stage of the model, we prevent unwanted features from propagating through the model. Moreover, since our intervention is *permanently* integrated into the generative backbone, rather than *non-permanently* operating externally on the text encoder as in Poppi et al. (2024), simply replacing the text encoder will not undo concept removal with our method. Unlike stand-alone SAE-based modules (Cywiński & Deja, 2025), our approach embeds the transcoder as a permanent, non-removable, integral component, ensuring persistent concept removal and robustness, even in local model deployments. Additionally, unlike SAeUron that intervenes in the middle of the network and requires extensive inference over multiple DM generation time-steps to collect training samples (Cywiński & Deja, 2025), our approach operates at the model input, where training data can be obtained directly from the encoding stage with minimal computation, both for DMs and IARs.

Finally, despite being located early in the model architecture, our method does not need a computationally expensive back-propagation through the large generative model. Instead, it can be trained stand-alone and separately from the image-generative-model to learn sparse monosemantic features, and then simply "plugged in". During the intervention, we only need to identify the features responsible for our target concept $t$, and redirect them to prevent the generation of $t$. We detail the training and the intervention for concept removal in the next sections.

### 4.2. Transcoder Training

In the training phase, the transcoder is optimized to replicate the behavior of the original transformation while imposing a monosemantic decomposition of concepts.

**Training Data Curation.** The curation of the transcoder's training data is lightweight and simple. It requires a set of prompts, for example, taken from open source datasets. These prompts are forwarded through the model's initial text encoder to yield the training data inputs for the transcoder. The ground truth outputs for the transcoder's training are obtained by inferring these embeddings through the model's original text transformation, *e.g.,* the MLP layer in DMs. Crucially, unlike SAeUron (Cywiński & Deja, 2025), our approach avoids the need for expensive full-model inference to gather activations. Based on our generated input-output pairs, the transcoder is then trained to approximate the original transformation.

**Training Objective.** The $\ell_0$ pseudo-norm measures the number of non-zero elements in a vector, which ideally we would like to minimize for the latent representation $\mathbf{z} \in \mathbb{R}^{m \cdot d}$ in a transcoder to enforce sparsity, but direct optimization with respect to this norm is challenging due to its non-convexity. Therefore, transcoder employs the $\ell_1$ norm as a convex surrogate, which promotes sparsity while enabling more efficient optimization. We observed that the original transcoder training with $\ell_1$ regularization introduces two key issues for effective concept removal. 1) First, $\ell_1$ is only an imperfect surrogate for the desired $\ell_0$ constraint and it only biases positive activations toward zero. The shrinkage of activations in the $\ell_1$ norm instead of zero-ing them out as in $\ell_0$ norm, prevents a clean separation between active and inactive units. 2) Second, the formulation often results in many "dead" latents that never activate, leading to under-utilization of the available latent space. For concept removal, both limitations are critical: the lack of a sharp boundary between concept-specific latents hinders precise isolation, thereby affecting both the target concept and unrelated concepts alike, leading to sub-optimal trade-offs. At the same time, the presence of dead latents reduces the model's capacity to encode distinct concepts, resulting in less removal capacity.

As a solution, we replace the ReLU with a TopK activation function, which enforces sparsity by retaining only the $k$ largest entries of the encoded representation. This provides a direct $\ell_0$ constraint rather than a surrogate. Our modified encoder–decoder mapping is given by

$$\begin{aligned} \mathbf{z} &= \mathrm{TopK}(W_{\mathrm{enc}}\mathbf{x} + \mathbf{b}_{\mathrm{enc}}), \\ \hat{\mathbf{y}} &= W_{\mathrm{dec}}\mathbf{z} + \mathbf{b}_{\mathrm{dec}.} \end{aligned} \tag{3}$$

Since the TopK operator is non-differentiable, we employ a straight-through estimator to propagate gradients during backpropagation. Additional implementation and optimization details are provided in Appendix Section E.1.

Inspired by work on training SAEs (Gao et al., 2024), we then train with a composite loss $\mathcal{L}$ consisting of three com-

ponents:

$$\mathcal{L} = \mathcal{L}_{\text{fidelity}} + \alpha_1 \, \mathcal{L}_{\text{multi-topK}} + \alpha_2 \, \mathcal{L}_{\text{aux}}, \qquad (4)$$

where $\mathcal{L}_{\text{fidelity}} = \|\mathbf{y} - \hat{\mathbf{y}}\|_2^2$ enforces alignment between the original transformation and the transcoder output. The multi-TopK loss, $\mathcal{L}_{\text{multi-topK}} = \|W_{\text{dec}}\mathbf{z}^{(4k)} + \mathbf{b}_{\text{dec}} - \mathbf{y}\|_2^2$, applies the same reconstruction objective with an expanded TopK budget (typically $4k$) to encourage progressive code utilization. Finally, the auxiliary loss, $\mathcal{L}_{\text{aux}} = \|W_{\text{dec}}\mathbf{z}^{(\text{aux})} + \mathbf{b}_{\text{dec}} - \mathbf{y}\|_2^2$, computes the fidelity objective using only inactive latents, thereby mitigating the problem of dead units. Each term is normalized by the variance of the target activations, so all losses represent fractions of unexplained variance. A detailed description of the loss components is provided in Appendix E.2.

### 4.3. Transcoder-based Intervention

We perform the intervention for concept removal by identifying the features responsible for one or multiple target concepts and redirecting them to prevent the visual generation of the specified concepts.

**Feature Identification.** Formally, let $\mathbf{z}(p_i) \in \mathbb{R}^{m \cdot d}$ denote the latent vector obtained after applying the TopK encoder to a token $p_i$. We define the activation indicator

$$\delta_i(p_i) = \mathbf{1}\{z^{(i)}(p_i) \neq 0\}, \quad i \in \{1, \dots, m \cdot d\},$$

which specifies whether latent $i$ is active for token $p_i$. For a target concept $t$ with tokens $\{p_{t1}, \dots, p_{tk}\}$, $\mathcal{A}_t = \bigcup_{i=1}^{k} \mathcal{A}_{ti}$ denotes the latent subset that encodes $t$, *i.e.*, the indices for removal:

$$\mathcal{A}_{ti} = \{i \mid \delta_i(p_{ti}) = 1\}.$$

**Concept Removal.** Once the subset of latents associated with $t$ has been identified, the goal is to block their contribution to the text representation. Since each latent corresponds to a column of the decoder weight matrix $W_{\text{dec}}$, suppression can be implemented by directly modifying these columns.

We modify $W_{\text{dec}}$ such that the activations of $\mathcal{A}_t$ are redirected to reproduce the output of the empty token $\emptyset$. Let $\mathbf{z}_t \in \mathbb{R}^{md}$ and $\mathbf{z}_\emptyset \in \mathbb{R}^{md}$ denote the latent activations for the target concept and the empty token, respectively. Because the TopK operator produces exactly $k$ active latents per token, each $\mathcal{A}_{ti}$ and $\mathcal{A}_\emptyset$ have the same cardinality. We therefore define a pairing

$$\pi : \mathcal{A}_{ti} \to \mathcal{A}_\emptyset$$

as a bijection between these sets. Any bijection leads to the same result. In our implementation, we adopt the simplest strategy: latents are paired in order, so the first latent of $\mathcal{A}_{ti}$ is matched to the first of $\mathcal{A}_\emptyset$, the second to the second, and

so on. The decoder is then modified as

$$\forall i \in \{1, \dots, md\},$$

$$W_{\text{dec}}^{(i)} = \begin{cases} \frac{z_\emptyset^{(\pi(i))}}{z_t^{(i)} + \varepsilon} W_{\text{dec}}^{(\pi(i))} & \text{if } i \in \mathcal{A}_t, \\ W_{\text{dec}}^{(i)} & \text{otherwise,} \end{cases} \quad (5)$$

with $\varepsilon > 0$ a small constant for numerical stability. Under this substitution, the decoder contribution for $\mathbf{z}_t$ becomes

$$W_{\text{dec}}\mathbf{z}_t = \sum_{i \notin \mathcal{A}_t} W_{\text{dec}}^{(i)} z_t^{(i)} + \sum_{i \in \mathcal{A}_t} \frac{z_\emptyset^{(\pi(i))}}{z_t^{(i)} + \varepsilon} W_{\text{dec}}^{(\pi(i))} z_t^{(i)}$$

$$\approx \sum_{i \in \mathcal{A}_t} W_{\text{dec}}^{(\pi(i))} z_\emptyset^{(\pi(i))} = W_{\text{dec}}\mathbf{z}_\emptyset, \quad (6)$$

showing that the contribution of $\mathbf{z}_t$ is redirected to reproduce the output of the empty token, thereby removing the target concept while preserving a coherent representation.

## 5. Empirical Evaluation

### 5.1. Experimental Setup

**Models.** We mainly evaluate our method on SOTA text-to-image generative models from two major architectures, namely DMs and IARs. For **DMs**, we consider SD3.5 (Esser et al., 2024a), Flux (Labs, 2024). For **IARs**, we analyze Infinity-2B and Infinity-8B (Han et al., 2025). This selection ensures coverage of both architectures and multiple model scales to highlight our method's generality.

**Concept Removal Tasks and Benchmarks.** We evaluate our method on **style removal** and **object removal** from UnlearnCanvas (Zhang et al., 2024b). UnlearnCanvas provides a benchmark for object and style unlearning. Yet, the benchmark's style classifier is trained on SD1.5 generations and generalizes poorly to modern architectures such as SD3.5, Flux, and Infinity (below 6% accuracy, see Table 1). Therefore, we replace it with LLaVA-1.6-Vicuna-7B (Liu et al., 2023) as a unified zero-shot classifier; Table 2 reports LLaVA classification accuracy on unmodified baseline outputs. To assess reliability, we conducted a human alignment study in which two independent raters judged 100 generated images (10 per style) on whether they exhibited the intended style. The raters agreed on 89 images; LLaVA matched the consensus on 84 of those, yielding a 94.4% match rate. Per-style results are reported in Table 3, confirming that LLaVA provides a reliable evaluation signal across all ten styles. With this external classifier, we focus on ten visually distinct styles (Cartoon, Cubism, Winter, Pop Art, Ukiyoe, Impressionism, Byzantine, Van Gogh, Bricks, Watercolor), while retaining the object categories defined in the original benchmark.

**Metrics.** We assess *concept removal* using classification-based metrics together with distributional quality measures.

*Table 1.* **UnlearnCanvas classifier accuracy on modern generative models.** Style accuracy is near-random across all four models, confirming that the original SD 1.5 classifier does not generalize.

| Model | Style Acc. (%) | Object Acc. (%) |
|---|---|---|
| SD3.5 | 3.96 | 93.58 |
| Flux | 2.12 | 90.64 |
| Infinity-2B | 5.32 | 95.32 |
| Infinity-8B | 6.06 | 95.04 |

*Table 2.* **LLaVA-based style and object classification accuracy on unmodified model outputs.** Style accuracy varies across architectures; Flux generates less style-consistent images, consistent with Zhang et al. (2025).

| Model | Style Acc. | Object Acc. |
|---|---|---|
| SD3.5 | 62.1% | 94.3% |
| Flux | 31.8% | 95.3% |
| Infinity-2B | 38.8% | 95.6% |
| Infinity-8B | 63.2% | 96.4% |

For a target concept $c$, we report **Unlearning Accuracy (UA)** as the fraction of images generated with prompts containing $c$ that are not classified as $c$. To measure preservation of unrelated content, we report two retention metrics: **In-domain Retain Accuracy (IRA)**, the classification accuracy on non-target concepts from the same domain (*e.g.,* other objects when unlearning one object), and **Cross-domain Retain Accuracy (CRA)**, the accuracy on concepts from a different domain (*e.g.,* style accuracy when unlearning an object).

*Image quality* is evaluated with the **Fréchet Inception Distance (FID)** (Heusel et al., 2017), **HPSv3** (Ma et al., 2025b), and **Aesthetics v2.5**[2] computed between intervened and original generations after excluding samples of the removed concept, and the **CLIPScore** (Hessel et al., 2021) under the same exclusion, as concept removal is supposed to break the alignment between prompts containing the target concept and their generated image.

**Transcoder.** We initialize the transcoder with an expansion factor of 16 and a TopK budget of $k = 32$. We train it for 100 epochs. We use the curated dataset from Cywiński & Deja (2025): 80 prompts per object across 20 objects, augmented with 10 style variants and a no-style variant, yielding 17,600 prompts in total. A complete description of hyperparameters and training settings is provided in Section E.1.

**Baselines.** We compare our approach against various state-of-the-art baselines. For the state-of-the-art models, namely SD3.5, Flux, Infinity-2B, and Infinity-8B, we compare BLOCK against UCE (Gandikota et al., 2024) and LOCOEDIT (Basu et al., 2024b), the most relevant white-

[2]https://github.com/discus0434/aesthetic-predictor-v2-5

*Table 3.* **Human alignment of LLaVA style classification.** For each style we report the number of images (out of 10) labeled by each rater as exhibiting the style, the ground-truth count (both raters agreed), the number correctly classified by LLaVA, and the resulting match rate.

| Style | Rater 1 | Rater 2 | Ground Truth | LLaVA | Match (%) |
|---|---|---|---|---|---|
| Van Gogh | 9 | 9 | 9 | 8 | 88.9 |
| Picasso | 9 | 8 | 8 | 7 | 87.5 |
| Cartoon | 10 | 10 | 10 | 10 | 100.0 |
| Cubism | 9 | 10 | 9 | 9 | 100.0 |
| Winter | 8 | 8 | 8 | 8 | 100.0 |
| Pop Art | 9 | 9 | 9 | 8 | 88.9 |
| Ukiyoe | 10 | 9 | 9 | 8 | 88.9 |
| Impressionism | 9 | 8 | 8 | 8 | 100.0 |
| Byzantine | 10 | 10 | 10 | 9 | 90.0 |
| Bricks | 9 | 10 | 9 | 9 | 100.0 |
| Total | 92 | 91 | 89 | 84 | 94.4 |

box editing methods that are directly applicable to both DMs and IARs. UCE has demonstrated SOTA unlearning accuracy (UA) in Zhang et al. (2024b), while LOCOEDIT is designed to perform localized interventions that aim to preserve model utility by restricting edits to concept-relevant layers. For SD3.5 and Flux, which use dynamic text representations that are incompatible with LOCOEDIT's original localization step, we adapt the method using zero-ablation as described in Section B.2. We further include architecture-specific baselines, namely EraseAnything (Gao et al., 2025) on SD3.5 and SafetyGap (Zhong et al., 2026) on Infinity-2B and Infinity-8B. For SD3.5, we additionally evaluate against SAeUron (Cywiński & Deja, 2025), AGE (Bui et al., 2025), ConceptPrune (Chavhan et al., 2025), GLoCE (Lee et al., 2025), and AdaVD (Wang et al., 2025). Full results are in Section F.1; BLOCK outperforms all methods.

### 5.2. Our Method Yields Effective Concept Removal

We first compare our method to the baselines on state-of-the-art models in Table 4. The results highlight that our method outperforms the baselines over all models. Concretely, our method delivers consistent improvements in concept removal across both style and object domains while maintaining high visual quality. On SD3.5, our approach achieves 69.6% UA, 67.3% IRA, and 92.6% CRA for style removal, outperforming LOCOEDIT and UCE by margins of 8-28%. For object removal on SD3.5, UCE attains a strong 84.5% UA, but our method achieves higher in-domain and cross-domain retention (93.0% IRA and 67.5% CRA), demonstrating that it more reliably removes the target concept while preserving fidelity to the original domain and maintaining transferability across domains.

On Flux, our method's improvements are even more pronounced. It reaches 88.6% UA and 96.4% CRA for style removal, with over 93% UA and 96.6% IRA for object removal, surpassing LOCOEDIT and UCE by wide margins. These results indicate that our approach is particularly effective at disentangling style attributes, which are often more

*Table 4.* **Style and object removal performance of the proposed method and SOTA baselines.** FID, CLIP, HPSv3, and Aesthetic v2.5 scores are reported on non-target data.

| Model | Method | Style Removal | | | Object Removal | | | FID ($\downarrow$) | CLIP ($\uparrow$) | HPSv3 ($\uparrow$) | Aesthetic v2.5 ($\uparrow$) |
|---|---|---|---|---|---|---|---|---|---|---|---|
| | | UA ($\uparrow$) | IRA ($\uparrow$) | CRA ($\uparrow$) | UA ($\uparrow$) | IRA ($\uparrow$) | CRA ($\uparrow$) | | | | |
| SD3.5 | LOCOEDIT | 41.49 | 59.23 | 84.57 | 37.87 | 85.78 | 58.89 | 54.66 | 0.3246 | 7.46 | 6.02 |
| | UCE | 61.78 | 61.89 | 83.47 | **84.45** | 82.23 | 60.93 | 62.34 | 0.3205 | 6.19 | 6.11 |
| | Ours | **69.60** | **67.30** | **92.60** | 68.00 | **93.00** | **67.50** | **51.89** | **0.3376** | **8.31** | **6.15** |
| Flux | LOCOEDIT | 66.45 | 33.23 | 83.44 | 35.75 | 91.23 | 35.76 | 55.56 | 0.2911 | 8.13 | 6.13 |
| | UCE | 67.43 | 34.78 | 76.56 | 87.45 | 82.34 | 36.43 | 58.90 | 0.2996 | 7.54 | 6.34 |
| | Ours | **88.60** | **36.10** | **96.40** | **93.20** | **96.61** | **38.20** | **51.67** | **0.3059** | **8.49** | **6.49** |
| Infinity-2B | LOCOEDIT | 81.27 | 37.65 | 62.45 | 85.72 | 68.09 | 34.52 | 91.55 | 0.3007 | 6.57 | 5.27 |
| | UCE | 68.59 | 35.09 | 74.29 | 72.48 | 69.34 | 31.43 | **77.56** | 0.3078 | 6.38 | 5.54 |
| | Ours | **86.80** | **39.60** | **90.30** | **91.20** | **84.50** | **38.30** | 90.92 | **0.3093** | **6.98** | **5.61** |
| Infinity-8B | LOCOEDIT | 80.07 | 59.61 | 63.47 | 88.67 | 70.54 | 57.44 | 56.21 | 0.3225 | 7.87 | 6.47 |
| | UCE | 76.75 | 58.76 | 68.67 | 75.35 | 69.41 | 56.56 | 62.34 | 0.3289 | 7.96 | 6.44 |
| | Ours | **85.40** | **63.70** | **95.50** | **90.20** | **95.70** | **61.40** | **53.93** | **0.3338** | **8.54** | **6.51** |

diffuse and challenging to isolate, while still retaining high accuracy within and across domains.

For Infinity-2B, LOCOEDIT performs reasonably well on object removal, yet our method still achieves the highest overall results, with 90.3% CRA for style removal and 91.2% UA for object removal. Compared to UCE, which struggles on this model, our method provides clear advantages across all three metrics. On the larger Infinity-8B model, our method again delivers the best performance, reaching 95.5% CRA for style removal and 95.7% IRA for object removal, confirming that our approach scales effectively with model size while ensuring both in-domain and cross-domain consistency across novel SOTA models and training paradigms.

In addition to removal accuracy, our approach preserves generative quality, as reflected by lower FID and higher CLIP scores across most settings. This balance between strong concept suppression and high-fidelity image synthesis highlights the effectiveness of our method compared to existing baselines, which often face trade-offs between removal strength and image quality. We complement these quantitative results with qualitative examples in Section C. Furthermore, the comparison of training time, storage, and memory requirements in Section 5.8 demonstrates a clear advantage of our method over training-based approaches.

### 5.3. Multi-Concept Removal

We assess our method's ability to remove multiple concepts in comparison to the baselines. Concretely, we distinguish between two setups. **1) Simultaneous Removal** removes $N$ concepts by identifying the corresponding latent subsets in the transcoder and jointly redirecting their decoder contributions to the empty token, ensuring that none of the $N$ concepts are expressed during the generation. **2) Sequential Removal** removes one concept at the time until reaching $N$. For our method, both setups are functionally identical

since after N sequential steps the same set of latents has been redirected as in the simultaneous case. In contrast, for baselines LOCOEDIT and UCE, sequential removal requires iterative weight edits, which accumulate across steps and lead to degraded performance compared to the simultaneous setting. For simultaneous removal, we re-run the experiment with different $N \in \{1, \ldots, 10\}$, for sequential removal, we continue the experiment from the removal of $N = 1$, and then add another one at the time, until we also reach $N = 10$.

We present the results for style removal in Figure 3, where the $y$-axis reports the average score defined as $(\text{UA} + (\text{IRA} + \text{CRA})/2)/2$, following Cywiński & Deja (2025). This score combines the unlearning and retain capabilities into a single metric for simpler comparison. We report the mean computed 5 random seeds. Similar results for object removal can be found in Figure 8 in the Appendix. Across all models and both setups, our method consistently achieves higher average scores than LOCOEDIT and UCE. In the sequential setting, the performance of the baselines drops sharply as more concepts are removed, reflecting the accumulation of errors from repeated weight edits. In contrast, our method remains stable, showing only a gradual decline even when ten concepts are removed. In the simultaneous setting, all methods perform better at small $N$, but the gap between our method and the baselines widens as $N$ increases, with our approach maintaining substantially higher scores. The advantage is especially pronounced for larger models such as SD3.5 and Infinity-8B, where the baselines degrade most quickly. These results demonstrate that our BLOCK provide robust multi-concept removal that scales reliably with $N$, avoiding the compounding errors of sequential editing while maintaining strong retention across both setups. While the results in Figure 3 are for style removal, the same trends can be observed for object removal as we show in Section F.2.

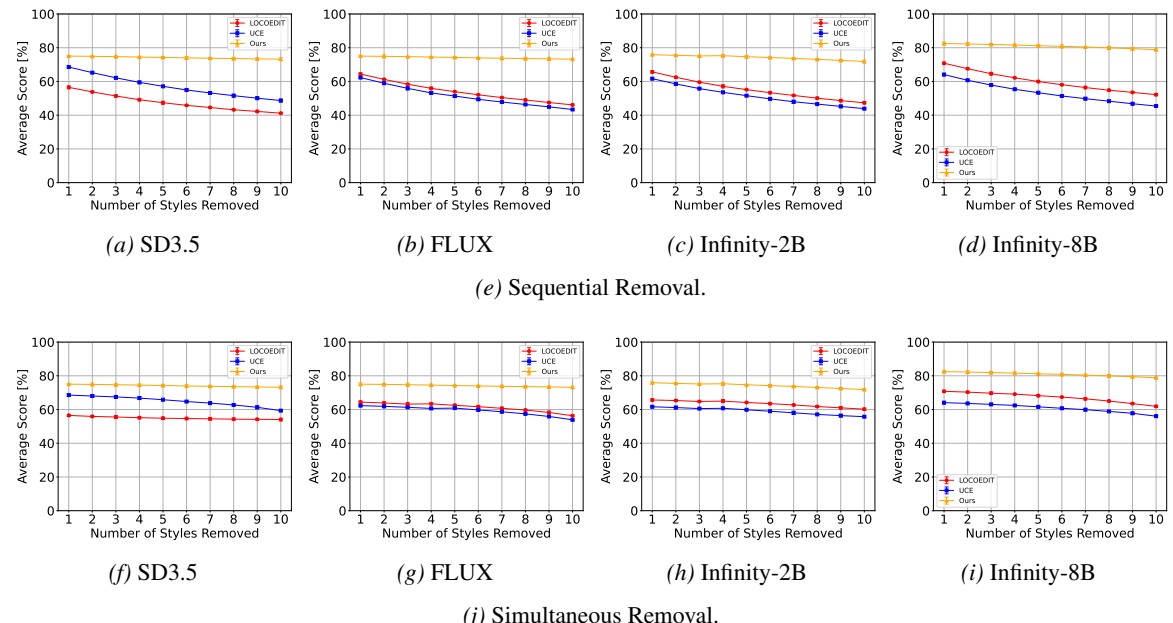

*(e)* Sequential Removal.

*(f)* SD3.5      *(g)* FLUX      *(h)* Infinity-2B      *(i)* Infinity-8B

*(j)* Simultaneous Removal.

*Figure 3.* **Multi-Concept Style Removal of Our Method and SOTA Baselines.**

*Table 5.* **Success Rates for the Ring-A-Bell Attack.**

| Method | SD3.5 | Flux | Inf-2B | Inf-8B |
|---|---|---|---|---|
| Original | 58.08 | 28.43 | 36.46 | 56.34 |
| LOCOEDIT | 57.36 | 26.32 | 24.67 | 22.61 |
| UCE | 43.44 | 24.45 | 25.31 | 24.43 |
| **Ours** | **25.21** | **15.62** | **21.23** | **19.78** |

*Table 6.* **MMA Diffusion and UnlearnDiff Attack Success Rates.**

| Method | SD3.5 | | Flux | |
|---|---|---|---|---|
| | MMA ↓ | UnlearnDiff ↓ | MMA ↓ | UnlearnDiff ↓ |
| Original | 61.48 | 63.74 | 30.11 | 29.19 |
| LOCOEDIT | 60.03 | 61.22 | 28.63 | 23.32 |
| UCE | 55.44 | 52.76 | 29.91 | 26.11 |
| **Ours** | **37.21** | **34.69** | **18.89** | **12.32** |

## 5.4. Robustness Evaluation

To evaluate the behavior of BLOCK under adversarial prompting, we adapt Ring-A-Bell (Tsai et al., 2024), an adversarial input-optimization framework originally proposed for SD1.4 to both SOTA DMs and IARs. Ring-A-Bell constructs adversarial queries by extracting semantic representations of sensitive concepts (e.g., nudity, violence, or artistic styles) from text encoders such as CLIP and then applying discrete optimization to craft prompts that attempt to reintroduce the suppressed concept. Importantly, the attack operates in a black-box mode and does not rely on access to model internals.

Table 5 shows Ring-A-Bell attack success rates (%) across models, with lower values indicating greater robustness. The original models are the most vulnerable, reaching 58.08% on SD3.5 and 56.34% on Infinity-8B. Both LOCOEDIT and UCE reduce these rates, but their protection remains partial, especially on SD3.5. Our approach achieves the lowest success rates across all four models, cutting attack success by 33 percentage points on SD3.5 (from 58.08% to 25.21%) and by over 36 points on Infinity-8B (from 56.34%

to 19.78%). Even on Flux and Infinity-2B, where baseline vulnerabilities are lower, our method still outperforms by clear margins.

Finally, we also ran additional adversarial evaluations based on diffusion-specific attack frameworks, namely MMA-Diffusion (Yang et al., 2024a) and Unlearn-Diff (Zhang et al., 2024c). The detailed description of the setup is presented in Appendix F.4. Our evaluations on the SOTA DMs SD3.5 and Flux in Table 6 reveal that BLOCK again achieves lower attack success rates than the existing baselines.

## 5.5. Evaluations Beyond Style and Object Removal

We further evaluate safety concept removal beyond style and object unlearning. On the Inappropriate Image Prompt (I2P) benchmark (Schramowski et al., 2023) for nudity removal on Flux, Table 7 shows that BLOCK achieves the lowest total detection count while maintaining competitive image quality. On the P4D red-teaming benchmark (Chin et al., 2023) on SD3.5, Table 8 shows that BLOCK achieves the lowest attack success rate among all methods.

*Table 7.* **I2P nudity removal on Flux.** Inappropriate image counts (lower is better) across three categories and total, together with FID. BLOCK achieves the lowest total count.

| Method | Common (↓) | Female (↓) | Male (↓) | Total (↓) | FID (↓) |
|---|---|---|---|---|---|
| Flux.1 [DEV] | 406 | 161 | 38 | 605 | 21.32 |
| LOCOEDIT | 239 | 72 | 24 | 335 | 26.68 |
| UCE | 122 | 39 | 12 | 173 | 30.71 |
| EraseAnything | 129 | 48 | 22 | 199 | 21.75 |
| **Ours** | **108** | **35** | **10** | **153** | 22.78 |

*Table 8.* **P4D attack success rate (%) on SD3.5.** Lower is better. BLOCK achieves the lowest attack success rate.

| Method | Attack Success Rate (%) (↓) |
|---|---|
| Original | 67.34 |
| LOCOEDIT | 58.78 |
| UCE | 41.15 |
| **Ours** | **35.61** |

### 5.6. Generalization to Out-of-Distribution Concepts

We verify that BLOCK generalizes to concepts absent from the transcoder's training set. Training on 5 of the 10 styles and evaluating on all 10, BLOCK achieves 52.63% UA on unseen styles with CRA above 90%, demonstrating that the sparse bottleneck representation captures transferable concept structure. In-distribution removal reaches 71.18% UA, confirming that broader training coverage further improves performance. Results are reported in Table 9.

*Table 9.* **Generalization to out-of-distribution styles on SD3.5.** The transcoder is trained on 5 styles and evaluated on all 10. In-distribution and out-of-distribution splits correspond to the seen and unseen subsets, respectively.

| Setup | UA (↑) | IRA (↑) | CRA (↑) |
|---|---|---|---|
| All data | 60.39 | 58.41 | 91.28 |
| In-distribution | 71.18 | 69.57 | 92.51 |
| Out-of-distribution | 52.63 | 51.03 | 90.88 |

### 5.7. Ablation on Bijection Pairing

The concept removal step in BLOCK relies on a bijection $\pi : \mathcal{A}_{ti} \rightarrow \mathcal{A}_{\emptyset}$ that maps each active latent of the target concept to one of the empty token. While any bijection is theoretically equivalent, we ablate whether the specific pairing is a sensible choice by comparing our default ordered pairing against random pairing across all four models in Table 10. Differences are below 0.1 percentage points on every metric, confirming that the pairing choice does not affect outcomes.

### 5.8. Efficiency

Table 11 compares BLOCK against UCE and LOCOEDIT in terms of computational overhead across four models. BLOCK requires less peak memory and less storage than

*Table 10.* **Effect of bijection pairing on concept removal performance.** Differences between ordered and random pairing are consistently below 0.1 percentage points, confirming that the choice of bijection does not affect outcomes.

| Model | Pairing | Style Removal | | | Object Removal | | |
|---|---|---|---|---|---|---|---|
| | | UA (↑) | IRA (↑) | CRA (↑) | UA (↑) | IRA (↑) | CRA (↑) |
| SD3.5 | Ordered | 69.60 | 67.30 | 92.60 | 68.00 | 93.00 | 67.50 |
| | Random | 69.57 | 67.33 | 92.55 | 67.96 | 93.06 | 67.45 |
| Flux | Ordered | 88.60 | 36.10 | 96.40 | 93.20 | 96.61 | 38.20 |
| | Random | 88.58 | 36.14 | 96.37 | 93.18 | 96.59 | 38.19 |
| Infinity-2B | Ordered | 86.80 | 39.60 | 90.30 | 91.20 | 84.50 | 38.30 |
| | Random | 86.77 | 39.59 | 90.32 | 91.18 | 84.51 | 38.27 |
| Infinity-8B | Ordered | 85.40 | 63.70 | 95.50 | 90.20 | 95.70 | 61.40 |
| | Random | 85.40 | 63.69 | 95.48 | 90.21 | 95.68 | 61.39 |

*Table 11.* **Efficiency comparison across methods.** Memory and storage are peak increases over the base model. Upfront cost is the one-time investment per model (layer localization for LOCOEDIT, transcoder training for BLOCK). Per-target unlearning is the time to suppress one concept after the upfront stage. Per-image overhead is the added inference latency.

| Model | Method | Memory (GB) | Storage (GB) | Upfront Cost (s) | Per-target Unlearn (s) | Per-image Overhead (s) |
|---|---|---|---|---|---|---|
| SD3.5 | LOCOEDIT | 3.0 | 1.50 | 300 | 0.50 | 0 |
| | UCE | 8.0 | 4.00 | 5 | 0.50 | 0 |
| | BLOCK | 2.4 | 2.15 | 6060 | 0.661 | 0.60 |
| Flux | LOCOEDIT | 4.0 | 2.00 | 400 | 0.50 | 0 |
| | UCE | 10.0 | 5.00 | 5 | 0.50 | 0 |
| | BLOCK | 1.9 | 1.93 | 5428 | 1.020 | 0.43 |
| Infinity-2B | LOCOEDIT | 1.5 | 0.80 | 100 | 0.30 | 0 |
| | UCE | 4.0 | 2.00 | 3 | 0.30 | 0 |
| | BLOCK | 0.67 | 0.512 | 648 | 0.323 | 0.17 |
| Infinity-8B | LOCOEDIT | 2.0 | 1.00 | 150 | 0.30 | 0 |
| | UCE | 6.0 | 3.00 | 3 | 0.30 | 0 |
| | BLOCK | 0.75 | 0.704 | 753 | 0.317 | 0.20 |

UCE. Its one-time upfront cost is higher than the closed-form baselines, ranging from approximately 10 minutes for Infinity-2B to approximately 100 minutes for SD3.5; this investment is paid once per model and is independent of the number of concepts to be removed. The higher cost for DMs relative to Infinity models stems from the MLP-layer transcoder, which requires more training data due to the pooled nature of that representation (see Section 4). Once trained, per-target removal takes under one second, making BLOCK practical for large-scale deployment.

## 6. Conclusion

We introduced a practical and scalable framework for robust concept removal in frontier text-to-image generative models. By integrating concept removal directly into the model backbone via transcoders, BLOCK achieves permanent and efficient interventions. Our approach yields strong performance in removing unwanted concepts while preserving image fidelity and robustness to adversarial attacks, and it generalizes across training paradigms and multi-concept settings. These advances position BLOCK as a foundation for more controllable, safe, and responsible deployment of generative models.

## Impact Statement

In this work, we propose a practical method for removing concepts in frontier image generative models. The method works by replacing an architecture-mandated transformation from the text input to the backbone with a transcoder. By integrating this surrogate directly into the architecture, the concept removal is effective even in white-box and open-source model access scenarios. In the broader picture, this contributes to improving the trustworthiness and controllability of modern large-scale generative models. While the approach might not address all the form of safety in adversarial settings, and current dedicated attacks against frontier models are to be further developed, it offers a computationally efficient mechanism to restrict concepts in deployed frontier image-generative models.

## Acknowledgements

This work was supported by the German Research Foundation (DFG) within the framework of the Weave Programme under the project titled "Protecting Creativity: On the Way to Safe Generative Models" with number 545047250.

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

# A. Extended Background

## A.1. Diffusion and Image AutoRegressive Models

**DMs.** DMs generate images by learning to invert a forward noising process. In the forward process, an image $\mathbf{x}_0$ is gradually perturbed into Gaussian noise $\mathbf{x}_T \sim \mathcal{N}(0, I)$ using a variance schedule $\{\beta_t\}_{t=1}^{T}$. A neural network is trained to approximate the reverse conditionals $p_\theta(\mathbf{x}_{t-1} \mid \mathbf{x}_t)$, typically parameterized as noise prediction. At inference, generation begins from random noise and applies the learned reverse steps until an image is reconstructed. Large-scale text-to-image systems such as Stable Diffusion (Rombach et al., 2022) adopted U-Net backbones as the denoising network. These models operate in a latent space obtained from an autoencoder and rely on cross-attention to inject conditioning. A separate text encoder produces embeddings once at the start of sampling. These text embeddings remain fixed for the entire denoising trajectory, and the U-Net attends to them as static keys and values at multiple layers. This design achieves a good performance but restricts text conditioning to a fixed representation. Recent models replace U-Nets with transformer-based denoisers. Diffusion Transformers (DiTs) (Peebles & Xie, 2022) represent images as sequences of latent patches and model dependencies with standard transformer blocks. The Multi-Modal DiT (MMDiT) (Esser et al., 2024a) extends this formulation to text-to-image generation by processing image and text tokens jointly in the same transformer stack. In architectures such as SD3.5, and Flux, text embeddings are updated dynamically at every layer through multimodal attention, rather than being fixed once at the beginning. This design integrates conditioning more deeply and scales effectively with model size. Modern DMs therefore fall into two main categories: U-Net-based systems with static text embeddings, and DiT-based systems with dynamic multimodal conditioning. Our method is compatible with both families, independent of their choice of backbone or conditioning strategy.

**Image AutoRegressive Models** IARs model the joint distribution of an image by decomposing it autoregressively into a product of conditional probabilities over discrete tokens. An image $\mathbf{x}$ is first quantized into a sequence $\mathbf{z} = (z_1, \ldots, z_N)$ using a tokenzier, and the conditional distribution is expressed as

$$p_\theta(\mathbf{z} \mid \mathbf{y}) = \prod_{i=1}^{N} p_\theta(z_i \mid z_{<i}, \mathbf{y}),$$

where $\mathbf{y}$ denotes a text prompt. During training, the model observes the ground-truth tokens $\{z_i\}_{i=1}^{N}$ from the tokenizer and learns to maximize their likelihood under the predicted conditional distributions. The training objective is therefore the negative log-likelihood

$$\mathcal{L}_{\text{IAR}} = -\sum_{i=1}^{N} \log p_\theta(z_i \mid z_{<i}, \mathbf{y}).$$

Infinity (Han et al., 2025) extends autoregressive image generation by replacing fixed-size token vocabularies with a bitwise multi-scale residual quantizer (Zhao et al., 2024). An image is hierarchically decomposed into binary variables across scales, and the model predicts each bit autoregressively given the previously generated bits and the text prompt. The training loss is

$$\mathcal{L}_{\text{Infinity}} = -\sum_{s=1}^{S} \sum_{b=1}^{B} \log p_\theta(z_{s,b} \mid z_{<s,<b}, \mathbf{y}),$$

where $s$ indexes scales and $b$ indexes bits within each scale. Text conditioning is provided by embeddings from a Flan-T5 encoder. These embeddings are computed once and remain fixed throughout generation, and are injected into the transformer backbone via cross-attention at every layer so that image tokens repeatedly attend to the same text features. Infinity therefore trains under a maximum likelihood objective while scaling autoregressive modeling efficiently to high-resolution text-to-image generation.

## A.2. Concept Editing/Removal

We present a taxonomy on concept removal approaches in Table 12.

### A.2.1. EXTERNAL METHODS

Several methods suppress target concepts without modifying model weights, intervening only externally at inference time.

**SLD** (Schramowski et al., 2023) (Safe Latent Diffusion) modifies the classifier-free guidance (CFG) formulation applied during denoising. In the standard setup, CFG interpolates between unconditional and conditional predictions. SLD augments

*Table 12.* **Taxonomy of concept removal methods.** Properties are marked with ✓ (yes) or ✗ (no).

| Method | Permanent | Paradigm-Agnostic | Fine-Tuning Free |
|---|---|---|---|
| FMN (Zhang et al., 2024a) | ✗ | ✗ | ✗ |
| MACE (Lu et al., 2024) | ✓ | ✗ | ✗ |
| MCE (Zhang et al., 2025) | ✓ | ✗ | ✗ |
| ESD (Gandikota et al., 2023) | ✓ | ✗ | ✗ |
| EA (Gao et al., 2025) | ✓ | ✗ | ✗ |
| CA (Kumari et al., 2023) | ✓ | ✗ | ✗ |
| EDiff (Wu et al., 2025) | ✓ | ✗ | ✗ |
| RECELER (Huang et al., 2023) | ✓ | ✗ | ✗ |
| SalUn (Fan et al., 2024) | ✓ | ✓ | ✗ |
| SHS (Wu & Harandi, 2024) | ✓ | ✓ | ✗ |
| SLD (Schramowski et al., 2023) | ✗ | ✓ | ✓ |
| SAFREE (Yoon et al., 2025) | ✗ | ✓ | ✓ |
| SEOT (Li et al., 2024) | ✗ | ✓ | ✗ |
| SAeUron | ✗ | ✓ | ✓ |
| SPM (Lyu et al., 2024) | ✓ | ✗ | ✗ |
| LOCO EDIT (Basu et al., 2024a) | ✓ | ✓ | ✓ |
| UCE (Gandikota et al., 2024) | ✓ | ✓ | ✓ |
| BLOCK (**Ours**) | ✓ | ✓ | ✓ |

this process with additional "safe" and "unsafe" guidance terms and a custom dynamic. This approach is appealing because it requires no retraining, but it incurs computational overhead due to the extra predictions. Moreover, constraining the CFG direction often reduces fidelity and prompt adherence. As an inference-time intervention, it can also be trivially disabled in local deployments.

**SAeUron** (Cywiński & Deja, 2025) introduces sparse autoencoders (SAEs) trained to disentangle semantic features in the residual stream of frozen DMs. The SAEs identify features corresponding to unwanted concepts, which can then be suppressed by blocking their activations before they re-enter the model. This approach provides finer control than global steering, but it relies on auxiliary modules trained outside the model. Because the underlying weights remain unchanged, the suppression can be undone simply by removing the SAE modules, making the intervention non-persistent.

**SEOT** (Li et al., 2024) suppresses unwanted content by directly optimizing text embeddings at inference. It first applies soft-weighted regularization to reduce redundant encodings of the negative concept in the embedding space. Then, during generation, it performs embedding optimization with two explicit losses: a suppression loss that discourages attention to the target concept and a preservation loss that encourages fidelity to the original prompt. This optimization is repeated for every input, which increases latency, and because it operates entirely in embedding space, the generative model itself remains unmodified.

**SAFREE** (Yoon et al., 2025) introduces a black-box filtering pipeline for unsafe content. It first identifies a toxic subspace in the text embedding representation, then projects prompt tokens orthogonally to that subspace, suppressing unsafe semantics before they reach the model. To maintain image quality, SAFREE further applies self-validating filtering and latent re-attention during denoising. While effective at filtering NSFW content, the method is entirely training-free and external to the model, which again makes it easy to bypass in local deployments.

**Receler** (Huang et al., 2023) introduces lightweight eraser modules that are attached to the cross-attention layers of frozen DMs. Instead of updating the model weights, Receler trains these add-on erasers with two objectives: a locality regularization that restricts modifications to features aligned with the target concept, and an adversarial prompt loss that

improves robustness to paraphrased or adversarially crafted prompts. This design enables concept suppression while preserving unrelated content more effectively than global steering methods. However, because the erasers are external components and the base model remains unchanged, the intervention is not persistent: users can simply remove the eraser modules to restore the original generation behavior. Furthermore, training erasers for each new concept still requires additional fine-tuning overhead, making the method less practical for multi-concept or sequential removal scenarios.

Overall, external methods avoid changing model weights, which makes them attractive when persistence is not required or when direct editing is impractical. However, because the base models remain unchanged, these interventions can be trivially removed, offering no lasting guarantee of concept removal. Some approaches operate purely at inference and add latency (SLD, SEOT, SAFREE), while others attach auxiliary modules that require additional training (SAeUron, Receler). In both cases, scalability to multi-concept or sequential removal is limited. By contrast, our method directly edits the model at the level of semantic features, providing persistent, interpretable, and generalizable concept removal across architectures.

### A.2.2. INTERNAL METHODS

A variety of model-editing methods have been proposed for concept removal. These approaches fall into two categories. *Training-based methods* adapt model parameters through fine-tuning or re-optimization. *Closed-form methods* instead modify attention weights through constrained projections. While many of these methods achieve strong results in specific settings, we do not include them as baselines. They rely on costly retraining, assume diffusion-specific architectures, or fail to generalize to autoregressive models. We now describe the most prominent approaches in detail.

**Forget-Me-Not** (Zhang et al., 2024a) proposes attention re-steering to suppress target concepts. It modifies cross-attention maps in U-Net-based DMs by fine-tuning them such that attention weights corresponding to a concept of interest are minimized, while unrelated tokens are preserved. Although Forget-Me-Not is relatively lightweight compared to full retraining, it still requires model fine-tuning for every new concept, which limits scalability to multi-concept and sequential removal scenarios. More critically, the method is inherently tied to the cross-attention mechanisms of U-Net denoisers and cannot transfer to architectures without these components, such as Transformer-based DMs (e.g., SD3.5 (Esser et al., 2024b)) or image autoregressive models such as Infinity (Han et al., 2025).

**Erased Stable Diffusion (ESD)** (Gandikota et al., 2023) erases concepts by leveraging classifier-free guidance. The method retrains the diffusion backbone so that generations conditioned on a target concept are pushed toward the generations of an "empty" prompt, thereby erasing the target concept during denoising. While effective for certain DMs, ESD is tightly coupled to the classifier-free guidance mechanism and the iterative denoising schedule of U-Net-based pipelines. ESD requires retraining for each concept, which is computationally expensive and unsuitable for large-scale or sequential removal tasks.

**EDiff** (Wu et al., 2025) formulates concept erasure as a bi-level optimization problem. At the inner level, it perturbs latent features associated with the target concept, while at the outer level it minimizes reconstruction losses to preserve unrelated content. This process requires gradient-based fine-tuning of the diffusion backbone, with optimization closely tied to the denoising schedules of U-Net-based DMs. Although effective within this scope, the reliance on iterative denoising and the computational demands of bi-level optimization make EDiff impractical for broader deployment. It cannot be applied to autoregressive models, and its high per-concept training cost prevents scalable or sequential removal.

**Safe-CLIP** (Poppi et al., 2024) fine-tunes CLIP encoders to reduce sensitivity to unsafe or undesired concepts. It trains on synthetic quadruplet datasets, which pair safe and unsafe samples along with positive and negative augmentations, in order to redirect unsafe embeddings into safer regions of the representation space while preserving semantic structure. This approach, however, requires large-scale curated training data and costly retraining for every new concept. More importantly, it is only applicable to models that rely on CLIP as their text encoder, such as early versions of Stable Diffusion. Modern SOTA DMs (e.g., SD3.5) and autoregressive models (e.g., Infinity) instead rely on encoders such as T5 in addition to CLIP, rendering Safe-CLIP incompatible with our setting.

**Minimalist Concept Erasure** (Zhang et al., 2025) formulates concept erasure as a direct optimization over generated outputs. The method minimizes the distributional distance between samples conditioned on a target concept and those from a neutral prompt, while simultaneously preserving unrelated content. Unlike prior approaches that intervene only at specific layers, Minimalist Concept Erasure backpropagates through the entire generation process. It further introduces learnable neuron masks that are optimized jointly with the model to suppress target concepts more robustly. Although effective, this approach requires end-to-end fine-tuning for each concept and is tied to diffusion-style generative dynamics, making it

computationally costly and inapplicable to autoregressive models.

In summary, the above methods illustrate diverse strategies for training-based erasure but share key limitations that make them unsuitable as baselines for our study. Training-based approaches such as Forget-Me-Not, ESD, Safe-CLIP, and Minimalist Concept Erasure require costly fine-tuning or retraining for each concept, which prevents scalable evaluation across multiple or sequentially added concepts. Some are further restricted to diffusion-specific architectures or text encoders, rendering them incompatible with modern DMs and autoregressive models such as Infinity. For these reasons, we do not include them as baselines in our evaluation. Instead, we focus on closed-form approaches, which directly edit cross-attention weights and are applicable across architectures without retraining.

### A.2.3. Closed-Form Editing Methods

Closed-form editing methods modify the linear projections inside cross-attention without any gradient-based retraining of the base model. They specify constraints for a set of edited concepts and a set of preserved concepts, then solve for new projection weights in closed form.

**UCE** (Gandikota et al., 2023) edits the key and value projection matrices $W_k$ and $W_v$ in cross-attention by enforcing linear constraints on specific text embeddings. Let $E = \{c_i\}$ be text embeddings of concepts to edit and let $P = \{c_j\}$ be embeddings of concepts to preserve. For each $c_i \in E$, a destination embedding $c_i^*$ is chosen and the desired target output is $v_i^* = W_{\text{old}} c_i^*$ for the projection under edit. UCE solves

$$\min_W \sum_{c_i \in E} \|W c_i - v_i^*\|_2^2 + \sum_{c_j \in P} \|W c_j - W_{\text{old}} c_j\|_2^2,$$

with closed-form solution

$$W = \left( \sum_{c_i \in E} v_i^* c_i^\top + \sum_{c_j \in P} W_{\text{old}} c_j c_j^\top \right) \left( \sum_{c_i \in E} c_i c_i^\top + \sum_{c_j \in P} c_j c_j^\top \right)^{-1}.$$

This update is applied to $W_k$ and $W_v$ in the chosen cross-attention layers. The method supports simultaneous multi-concept edits by stacking constraints for many $c_i$, and it ensures invertibility by augmenting the preservation set with canonical basis directions when needed. Erasure is implemented by choosing $c_i^*$ that maps to a neutral destination such as a generic token. The same formulation also covers debiasing and moderation by suitable choices of $c_i^*$. UCE generalizes TIME, and it is applicable to any architecture that uses linear cross-attention projections.

**LOCOEDIT** (Basu et al., 2024a) performs localized cross-attention edits by restricting updates to the layers and heads most responsible for representing a target concept. These are identified by contrasting activations from concept prompts against neutral prompts, which highlights the parameters most sensitive to the concept. For the selected $W_k$ and $W_v$ matrices, LOCOEDIT solves a regularized least-squares problem

$$\min_W \sum_{c_i \in E} \|W c_i - v_i^*\|_2^2 + \lambda \|W - W_{\text{old}}\|_2^2,$$

with closed-form solution

$$W = \left( \sum_{c_i \in E} v_i^* c_i^\top + \lambda W_{\text{old}} \right) \left( \sum_{c_i \in E} c_i c_i^\top + \lambda I \right)^{-1}.$$

Here, $E$ is the set of embeddings for concepts to edit, $v_i^*$ are their desired target projections, and $\lambda$ controls the strength of the update relative to the original weights $W_{\text{old}}$. By confining edits to concept-relevant regions, LOCOEDIT reduces interference with unrelated concepts while preserving training-free efficiency. Like UCE, it applies to any architecture with linear cross-attention projections and requires no retraining.

**MACE** (Lu et al., 2024) extends UCE to large-scale concept removal by training LoRA adapters for individual concepts and fusing them through a closed-form integration step. This hybrid design enables simultaneous erasure of up to 100 concepts but requires diffusion-specific denoising schedules and per-concept LoRA training, making it computationally expensive and inapplicable to autoregressive models.

In summary, closed-form editing methods offer efficient, retraining-free updates but vary in their ability to isolate target concepts without harming unrelated features. We include UCE and LOCOEDIT as baselines, since they are closed-form, architecture-agnostic, and applicable without retraining.

## B. Baseline Setup

### B.1. UCE

We implement UCE (Gandikota et al., 2023) following its original closed-form formulation. For each target concept, we construct the *edit set $E$* from text embeddings of the corresponding concept tokens. The *preservation set $P$* is sampled from unrelated tokens in the same vocabulary, ensuring that both object and style categories are represented. As in our method, the destination embedding $c_i^*$ is set to the embedding of the empty token, such that edited projections for target concepts map to a neutral representation.

The update is applied to the key and value projection matrices $W_k$ and $W_v$ in all cross-attention layers. Multi-concept removal is handled in two ways. For **simultaneous removal**, constraints for multiple concepts are stacked to compute a single joint update. For **sequential removal**, UCE is re-applied iteratively: after editing one concept, the updated weights are used as $W_{\text{old}}$ for the next edit, accumulating changes across concepts.

This setup is consistent across all models, including SD3.5, Flux, and Infinity, since the UCE formulation does not require architecture-specific retraining or model internals beyond cross-attention projections.

### B.2. LOCOEDIT

LOCOEDIT (Basu et al., 2024a) localizes concept-relevant layers by contrasting concept and neutral prompts. This procedure is well-suited to U-Net-based DMs, where the text embeddings remain fixed across all cross attention layers during denoising. In more recent architectures such as SD3.5 (Esser et al., 2024a) and FLUX (Labs, 2024) however, text and image features are updated jointly inside the multimodal transformer blocks, so the assumption of static text embeddings no longer holds. As a result, LOCOEDIT's original localization step is not directly applicable.

To adapt the method, we replaced LOCOEDIT's contrastive localization step with a zero-ablation procedure. We sampled 100 random prompts from the MS-COCO training set (Lin et al., 2014). For each prompt, we performed zero-ablation of the key and value projections across a sliding window of three consecutive cross-attention layers and measured the resulting CLIP score. The influence of each layer was quantified as the mean score drop over all prompts. We then selected the three layers with the largest drop as the localized set used for editing. For SD3.5, this setup identified attention layers 18, 19, and 20 as the most influential, while for Flux the most influential attention layers were 1, 2, and 3. These sets were then used in place of the original localization step for LOCOEDIT.

## C. Qualitative Results

We provide qualitative examples illustrating the effect of concept removal across models and baselines. For each category, we report the prompts and seed used to generate the images. Representative generations are shown in Figures 4 and 5, demonstrating the suppression of target styles and objects while retaining unrelated content.

## D. Extended Insights into our BLOCK Method

### D.1. Model-Architecture-Dependent Subtleties

Across the considered architectures, text representations enter the generative backbone through two possible network gates. The first, present in all models, is a linear projection layer that maps the text encoder outputs to the model dimension. In SD3.5 and Flux, there is an additional gate given by an MLP applied to a pooled text representation. Its output is combined with that of another MLP processing the noise schedule, and the sum is used for modulation throughout the network. Despite operating on a pooled representation, we found it necessary to intervene at this layer as well to ensure effective suppression of targeted concepts.

In Infinity models, a special first token is constructed from a pooled embedding of the text representation. This token is only used at the very beginning of the model and is not essential for downstream generation. We therefore omit it from our intervention, as experiments confirmed that transcoder substitution at the standard projection layer is sufficient.

# E. Extended Experimental Setup

We present further details of the experimental setup.

## E.1. Transcoder Training Details

**Training Data.** For the modulation MLP in SD3.5 and Flux, pooling collapses each prompt to a single vector, so we instead sample 30,000 prompts from MS-COCO (Lin et al., 2014) and apply the same style augmentation, resulting in 330,000 training pairs.

**Hyperparameters.** We set the sparsity level to $k = 32$ and the expansion factor to 16. Training uses the fused Adam optimizer with learning rate $\eta = 10^{-4}$, parameters $(\beta_1, \beta_2) = (0.9, 0.999)$, and stability constant $\epsilon = 6.25 \times 10^{-10}$. We apply an exponential moving average (EMA) of parameters with decay 0.999. Gradients are clipped to an $\ell_2$-norm of 1. We train each transcoder for 100 epochs, using a batch size of 16384 for projection-layer surrogates and 1024 for MLP-layer surrogates.

**Initialization.** When the input and output dimensions are equal, the decoder matrix is initialized as the transpose of the encoder. If they differ, we perform a QR decomposition of the encoder weights to obtain an orthogonal basis. If the resulting basis is smaller than the output dimension, we augment it with gaussian random vectors, apply QR again, and then truncate or pad to initialize the decoder. This procedure ensures orthogonality and prevents degenerate initialization.

**Efficiency.** We implement sparse–dense multiplication using the Triton kernel of Gao et al. (2024), reducing the forward-pass cost by approximately a factor of two. Each forward pass therefore requires only one large dense matrix multiplication, making training tractable even with an expansion factor of 16.

All training runs were conducted on NVIDIA A100 GPUs with 40GB of memory.

## E.2. Transcoder Loss Details

Let $(\mathbf{x}, \mathbf{y})$ denote an input–output pair of the original transformation. The encoder–decoder mapping is defined as

$$\mathbf{z} = \mathrm{TopK}(W_{\mathrm{enc}}\mathbf{x} + \mathbf{b}_{\mathrm{enc}}),$$
$$\hat{\mathbf{y}} = W_{\mathrm{dec}}\mathbf{z} + \mathbf{b}_{\mathrm{dec}}.$$

The composite training loss consists of three components:

**Fidelity loss.** This term enforces alignment between the transcoder output and the target activations of the original transformation:

$$\mathcal{L}_{\mathrm{fidelity}} = \frac{\|\hat{\mathbf{y}} - \mathbf{y}\|_2^2}{\mathrm{Var}(\mathbf{y})}. \tag{7}$$

**Multi-TopK loss.** To promote a smoother and more progressive allocation of importance across latent units, the fidelity objective is recomputed with an expanded TopK budget, typically $4k$. Let $\mathbf{z}^{(4k)}$ denote the latent vector obtained under this setting:

$$\mathcal{L}_{\mathrm{multi\text{-}topK}} = \frac{\|W_{\mathrm{dec}}\mathbf{z}^{(4k)} + \mathbf{b}_{\mathrm{dec}} - \mathbf{y}\|_2^2}{\mathrm{Var}(\mathbf{y})}. \tag{8}$$

**Auxiliary loss.** To mitigate the problem of *dead latents*—units that fail to activate—we keep track of activations using the multi-TopK setting. A latent is declared dead if it has not fired in the last $10^7$ forward passes. If at least one dead latent exists, an auxiliary loss is applied. Specifically, the fidelity objective is recomputed with a restricted TopK operator that is allowed to select only from the set of dead latents. The budget $k_{\mathrm{aux}}$ is set to the minimum between the number of dead latents and half of the input dimension. Let $\mathbf{z}^{(\mathrm{aux})}$ denote the resulting latent vector. The auxiliary loss is then

$$\mathcal{L}_{\mathrm{aux}} = \frac{\|W_{\mathrm{dec}}\mathbf{z}^{(\mathrm{aux})} + \mathbf{b}_{\mathrm{dec}} - \mathbf{y}\|_2^2}{\mathrm{Var}(\mathbf{y})}. \tag{9}$$

In summary, the composite loss in Eq. Equation (4) balances fidelity to the original transformation, progressive code utilization, and prevention of dead latents, with $\alpha_1 = \frac{1}{8}$ and $\alpha_2 = \frac{1}{32}$ in all our experiments.

*Table 13.* **Ablation of loss components on Infinity-2B for class unlearning.** We report UA, IRA, and CRA on the I2P benchmark, comparing different training configurations of our transcoder.

| Setting | Style Removal | | | Mean ($\uparrow$) |
|---|---|---|---|---|
| | UA ($\uparrow$) | IRA ($\uparrow$) | CRA ($\uparrow$) | |
| Infinity-2B (Fidelity only) | 88.60 | 25.40 | 59.87 | 57.96 |
| Infinity-2B (Fidelity + Top-$k$) | 86.80 | 26.57 | 74.72 | 62.69 |
| Infinity-2B (Fidelity + Aux) | 85.00 | 25.10 | 67.88 | 59.33 |
| Infinity-2B (Full) | 86.80 | 39.60 | 90.30 | 72.23 |

### E.3. Transcoder Loss Ablation

We conduct an ablation study on the loss components of our transcoder training. Specifically, we evaluate style unlearning performance on Infinity-2B under three configurations: **(i)** fidelity loss only, **(ii)** fidelity + multi–Top-$k$ loss, and **(iii)** fidelity + auxiliary loss. We do not evaluate the combination of multi–Top-$k$ and auxiliary loss, as it is effectively equivalent to the fidelity + auxiliary formulation with a larger effective $k$. The results are reported in Table 13.

### E.4. Llava Classifier

To evaluate concept removal performance, we require classifiers that can detect whether a generated image still exhibits the target concept. Following prior work, we adopt LLaVA-1.6-Vicuna-7B (Liu et al., 2023) as a zero-shot classifier, but adapt its usage to our benchmark settings.

**Style Classification.** For the style removal experiments, we prompt LLaVA with a fixed instruction asking it to assign an image to exactly one artistic style from a predefined list. UnlearnCanvas constructed a custom dataset by combining real images with stylization techniques and trained both Stable Diffusion 1.5 and a dedicated style classifier on it. However, this setup is tailored to SD 1.5 and its classifier, making it unsuitable for evaluating other architectures such as SD 3.5, Flux, and Infinity (Table 1). To ensure consistency across models, we instead employ LLaVA as a unified zero-shot classifier. We reduce the style set from 50 (as in UnlearnCanvas) to 10 categories to balance evaluation cost while maintaining coverage of diverse artistic modes. The exact LLaVA prompt used for style classification is shown in Figure 6.

**Object Classification.** For object removal experiments, we follow a similar procedure. LLaVA is instructed to classify each generated image into exactly one of 20 object categories, corresponding to the object set from UnlearnCanvas. This ensures a fair evaluation of whether removed objects still appear in generated samples, while also verifying that unrelated objects are retained. The full prompt for object classification is shown in Figure 7.

**Evaluation.** For both style and object removal, LLaVA returns an integer corresponding to the predicted class. Given a target concept $c$, we report:

- **Unlearning Accuracy (UA)**: Fraction of images generated with prompts containing $c$ that are not classified as $c$.

- **In-domain Retain Accuracy (IRA)**: Accuracy on non-target concepts from the same domain (e.g., other objects when unlearning one object).

- **Cross-domain Retain Accuracy (CRA)**: Accuracy on concepts from a different domain (e.g., style accuracy when unlearning an object).

This setup ensures that classification is consistent across all generative models considered, without requiring domain-specific classifiers or retraining. Table 2 reports the classification accuracy of the unmodified baseline models. Consistent with the findings of Zhang et al. (2025), we observe that Flux struggles to render artistic styles accurately.

## F. Additional Empirical Results

### F.1. Additional Baselines

To complement the cross-architecture comparison with UCE and LOCOEDIT in the main paper, we evaluate BLOCK against additional concept-removal methods, separately for SD3.5 and for Infinity-2B and Infinity-8B.

*Table 14.* **Comparison against additional recent baselines on SD3.5.** BLOCK achieves the highest score on every metric.

| Method | Style Removal | | | Object Removal | | |
|---|---|---|---|---|---|---|
| | UA (↑) | IRA (↑) | CRA (↑) | UA (↑) | IRA (↑) | CRA (↑) |
| EraseAnything | 62.54 | 64.49 | 87.32 | 65.32 | 88.21 | 64.36 |
| SAeUron | 66.54 | 62.41 | 87.47 | 64.45 | 90.11 | 62.31 |
| AGE | 60.18 | 57.09 | 81.47 | 60.75 | 78.96 | 57.43 |
| ConceptPrune | 52.19 | 50.94 | 76.42 | 57.81 | 72.34 | 48.32 |
| GLoCE | 58.28 | 55.32 | 84.88 | 61.78 | 84.51 | 59.34 |
| AdaVD | 63.71 | 60.11 | 85.41 | 61.44 | 86.61 | 60.49 |
| BLOCK | **69.60** | **67.30** | **92.60** | **68.00** | **93.00** | **67.50** |

*Table 15.* **Comparison against SafetyGap on Infinity-2B and Infinity-8B.** BLOCK improves over SafetyGap on every metric across both model sizes.

| Model | Method | Style Removal | | | Object Removal | | |
|---|---|---|---|---|---|---|---|
| | | UA (↑) | IRA (↑) | CRA (↑) | UA (↑) | IRA (↑) | CRA (↑) |
| Infinity-2B | SafetyGap | 71.27 | 24.88 | 83.56 | 83.18 | 78.44 | 26.88 |
| | BLOCK | **86.80** | **39.60** | **90.30** | **91.20** | **84.50** | **38.30** |
| Infinity-8B | SafetyGap | 76.61 | 57.33 | 90.78 | 86.41 | 89.39 | 52.21 |
| | BLOCK | **85.40** | **63.70** | **95.50** | **90.20** | **95.70** | **61.40** |

**Comparison on SD3.5.** We compare against EraseAnything (Gao et al., 2025), a training-based concept-erasure method designed for rectified-flow transformer diffusion models, alongside SAeUron (Cywiński & Deja, 2025), AGE (Bui et al., 2025), ConceptPrune (Chavhan et al., 2025), GLoCE (Lee et al., 2025), and AdaVD (Wang et al., 2025). These span both external (SAeUron) and internal (the remaining five) approaches. Table 14 reports the results. BLOCK achieves the highest score on every metric across all methods.

**Comparison on Infinity-2B and Infinity-8B.** "Closing the Safety Gap" (Zhong et al., 2026) (SafetyGap) is a concept-removal method targeting visual autoregressive models. We evaluate it on Infinity-2B and Infinity-8B in Table 15. BLOCK improves over SafetyGap on every metric across both model sizes, with the largest gains on Infinity-2B, where BLOCK improves Style IRA by 14.7 points and Object UA by 8 points.

### F.2. Multi-Concept Removal: Objects

As shown in Figure 8, across both sequential and simultaneous setups, our method consistently outperforms LOCOEDIT and UCE. In the sequential case, baselines degrade rapidly as more objects are removed. On SD3.5 and Infinity-8B, LOCOEDIT falls by more than 30 points as $N$ increases, and UCE shows a steady decline as well. Flux exhibits a similar trend, with both baselines losing around 15–20 points by the tenth removal. In contrast, our method shows only a gradual decrease across all models and sustains average scores above 70% even at $N = 10$, demonstrating robustness to accumulated interventions.

In the simultaneous case, where all concepts are removed in one step, performance for all methods is initially higher, but the gap widens with increasing $N$. On Flux and Infinity-2B, our approach maintains almost flat performance curves, while LOCOEDIT and UCE steadily lose accuracy. On the larger models, SD3.5 and Infinity-8B, the baselines degrade most quickly, whereas our method remains stable above 75%.

These findings confirm that transcoders provide reliable multi-concept removal across both diffusion and autoregressive models, scaling to larger $N$ without the sharp performance drops observed in prior methods.

## F.3. Additional Quality Metrics

To further assess the perceptual quality of edited images, we compute HPSv3 (Ma et al., 2025a) and Aesthetic v2.5[3] scores for all editing methods across the five model families evaluated in the main paper. For each model, we generate edited outputs using LOCOEDIT, UCE, and BLOCK under the same prompts used in our core experiments. We also include a No Edit reference, representing the original model output with no concept removal applied.

HPSv3 is used to measure prompt conditioned human preference quality, capturing realism, semantic alignment, and overall perceptual fidelity. Aesthetic v2.5 evaluates only visual appeal and stylistic quality, independent of prompt correctness. Each metric is computed directly on the generated images without additional filtering or post processing. All models are run with their recommended inference settings to ensure consistency and fairness in comparison.

Table 16 reports the full set of scores. BLOCK consistently remains closest to the No Edit reference across both metrics and all architectures, indicating that it achieves concept removal while retaining perceptual quality more effectively than LOCOEDIT and UCE. This behavior is also reflected in the qualitative examples shown in Figure 4 of the appendix.

*Table 16.* HPSv3 and Aesthetic v2.5 scores for all editing methods across modern model families. BLOCK remains closest to the No Edit reference on both metrics, indicating stronger preservation of perceptual and aesthetic quality.

| Method | HPSv3 ↑ | | | | Aesthetic v2.5 ↑ | | | |
|---|---|---|---|---|---|---|---|---|
| | SD3.5 | FLUX | Infinity-8B | Infinity-2B | SD3.5 | FLUX | Infinity-8B | Infinity-2B |
| No Edit (Base Model) | 8.63 | 8.87 | 8.61 | 7.13 | 6.23 | 6.77 | 6.58 | 5.88 |
| LOCOEDIT | 7.46 | 8.13 | 7.87 | 6.57 | 6.02 | 6.13 | 6.47 | 5.27 |
| UCE | 6.19 | 7.54 | 7.96 | 6.38 | 6.11 | 6.34 | 6.44 | 5.54 |
| BLOCK | **8.31** | **8.49** | **8.54** | **6.98** | **6.15** | **6.49** | **6.51** | **5.61** |

## F.4. Additional Robustness Results

To evaluate BLOCK under a broader range of adversarial conditions, we include additional benchmarks based on MMA Diffusion (Yang et al., 2024a) and UnlearnDiff (Zhang et al., 2024c), two attack frameworks designed to recover suppressed concepts through optimized prompts. The results in Table 6 show clear differences across methods on SD3.5 and Flux. LOCOEDIT consistently yields the highest attack success rates, and UCE exhibits a similar vulnerability pattern. In contrast, BLOCK achieves the lowest success rates for both attacks on both architectures, with the largest margin appearing on Flux, indicating greater resilience to adversarial prompting in models with stronger text–backbone bottlenecks. While these evaluations do not constitute comprehensive robustness guarantees, they provide evidence that integrating the intervention into the model's mandatory text-conditioning layer can improve resistance to targeted attempts to reintroduce removed concepts.

## G. Limitations

BLOCK requires a one-time transcoder training pass per model. Although this cost is modest in absolute terms (approximately 10 to 100 minutes depending on model size), it must be repeated when a new backbone is deployed. Concept removal itself, however, requires no retraining once the transcoder is in place.

---

[3]https://github.com/discus0434/aesthetic-predictor-v2-5

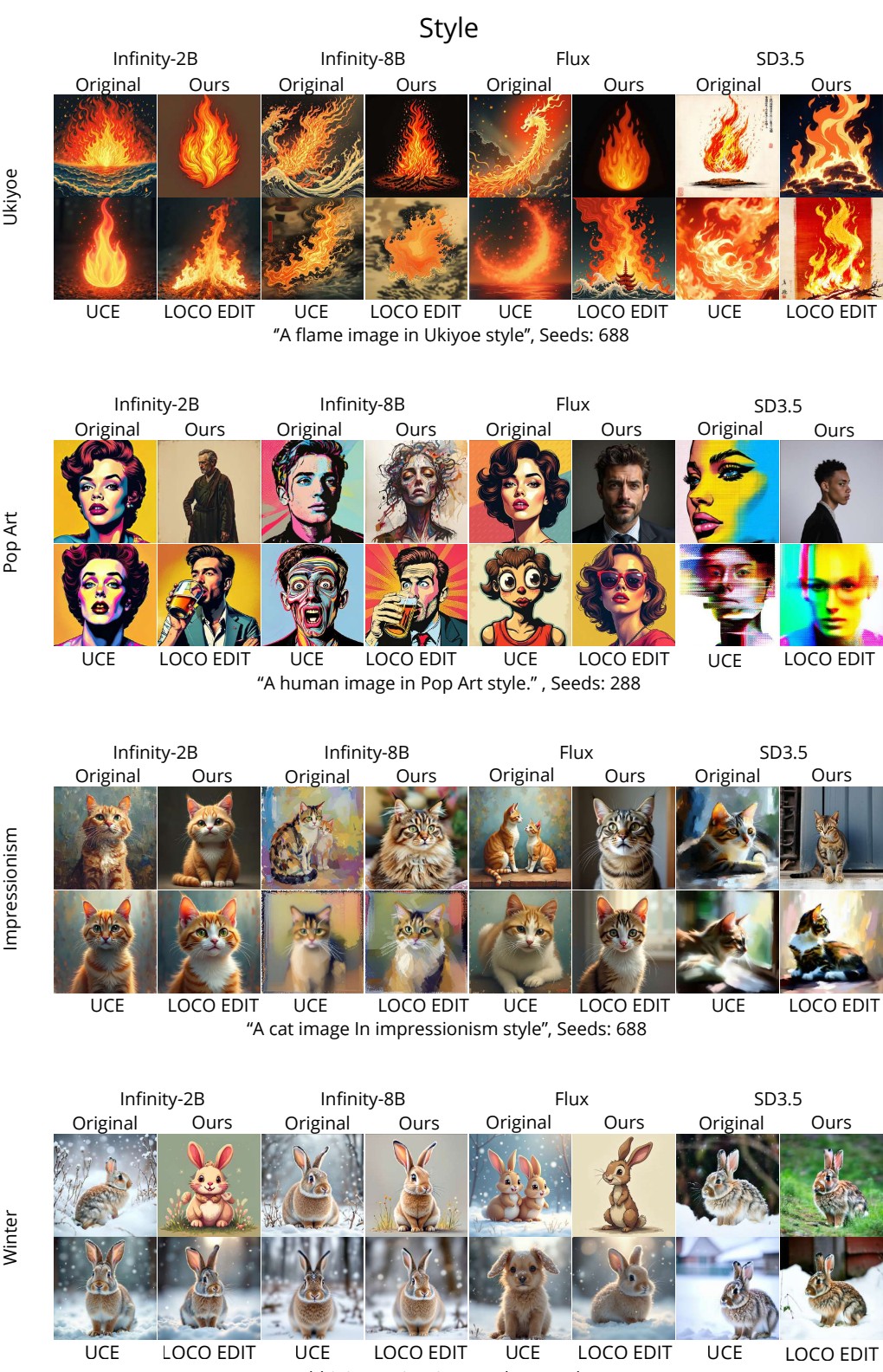

*Figure 4.* Qualitative results for style removal across models and baselines. Prompts and seeds are listed below each image.

## Object

*Figure 5.* Qualitative results for object removal across models and baselines. Prompts and seeds are listed below each image.

You are an image classifier. Classify the artistic style of the given image.

**Instruction:** Choose exactly **one** option from the numbered list below. Respond with **only the number**.

**Options:**

1. Van Gogh

2. Picasso

3. Cartoon

4. Cubism

5. Winter

6. Pop Art

7. Ukiyoe

8. Impressionism

9. Byzantine

10. Bricks

*Figure 6.* Prompt used for LLaVA-based style classification in our evaluation. The model must select exactly one label, ensuring consistent evaluation across generated samples.

Classify the object depicted in this image.
Choose exactly one option from the numbered list.
Respond with only the number.

**Object categories:**

1. Architecture

2. Bear

3. Bird

4. Butterfly

5. Cat

6. Dog

7. Fish

8. Flame

9. Flowers

10. Frog

11. Horse

12. Human

13. Jellyfish

14. Rabbits

15. Sandwich

16. Sea

17. Statue

18. Tower

19. Tree

20. Waterfalls

*Figure 7.* Prompt used for object classification in our LLaVA-based evaluation.

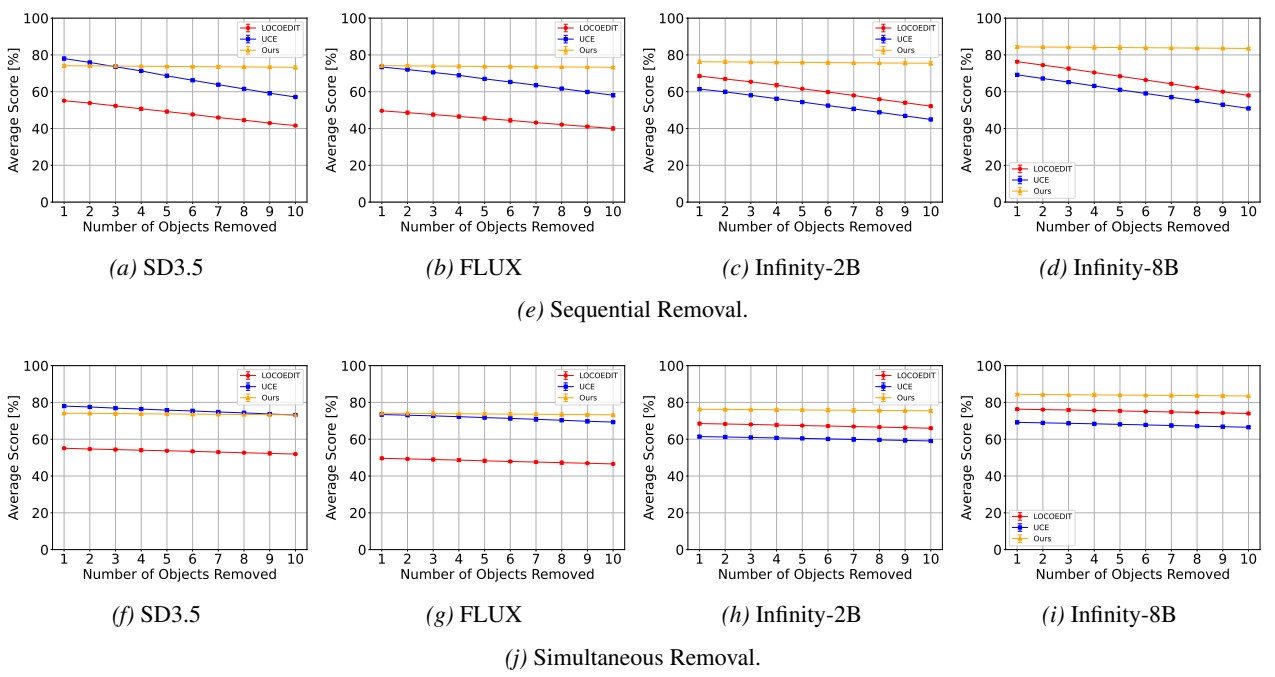

*(a)* SD3.5     *(b)* FLUX     *(c)* Infinity-2B     *(d)* Infinity-8B

*(e)* Sequential Removal.

*(f)* SD3.5     *(g)* FLUX     *(h)* Infinity-2B     *(i)* Infinity-8B

*(j)* Simultaneous Removal.

*Figure 8.* **Multi-Concept Removal of Our Method and SOTA Baselines for Objects.**

