# OpenReview forum: "Concept Removal for Frontier Image Generative Models"
_ICML.cc/2026/Conference — ICML 2026 regular_

### Official Review · Reviewer_Q43W · 2026-02-23

**Soundness:** 3
**Presentation:** 3
**Significance:** 2
**Originality:** 3
**Overall Recommendation:** 4
**Confidence:** 4

**Summary:**

This paper studies concept removal in large generative models and proposes an architecture-integrated approach based on sparse transcoder replacement. The authors introduce a sparse transformation module (e.g., transcoder) at the model’s bottleneck layer to approximate the original text-to-hidden mapping while enforcing Top-K sparsity to encourage feature disentanglement.
After training the transcoder to mimic the original transformation, concept-related latent features are identified and selectively modified to suppress specific concepts. Unlike external suppression modules, the proposed method performs in-place replacement within the model architecture, aiming to achieve more persistent and harder-to-bypass removal. Experiments demonstrate improved performance over two concept removal baselines, suggesting that architectural integration of sparse feature control can enhance representation-level concept suppression.

**Compliance With Llm Reviewing Policy:**

Affirmed.

**Final Justification:**

After considering both the paper and the authors’ response, my final assessment is more positive than my initial one. I found the work technically sound and clearly motivated from the start, with a practically important problem setting. My main concerns were the limited empirical comparison, the need for a clearer distinction from prior SAE-based representation-level removal methods, and the lack of sufficiently explicit discussion of robustness and limitations. The rebuttal addressed my concerns.

**Key Questions For Authors:**

Please refer to Weakness.

**Limitations:**

From my point of view, it would benefit from more systematic robustness analysis, such as testing stability under adaptive prompting, or potential bypass strategies. Such evaluation would strengthen confidence in the persistence and generality of the proposed removal mechanism.

**Strengths And Weaknesses:**

Strengths
The paper tackles an important and timely problem: concept removal in large generative models. The method is technically sound and clearly motivated. The idea of inserting a sparse transcoder into the bottleneck layer is practical and well explained. The in-place replacement design makes the removal more persistent than external modules. The experiments show consistent improvements over two baselines. The approach is easy to understand and appears implementable. While it builds on existing ideas in sparse feature decomposition, the architectural integration is creative and deployment-oriented.

Weakness
1. The empirical evaluation feels somewhat limited. The method is compared against only two recent baselines from 2024. A broader comparison would strengthen the paper. For example, including methods from different paradigms, such as loss-level unlearning, SAE-based representation suppression, or AGE, Concept Prune, would help clarify where this work stands in the overall concept removal landscape.

2. The overall idea feels somewhat incremental at a conceptual level. Sparse feature decomposition and selective suppression have already been explored in prior SAE-based removal work. The main difference here is architectural integration and in-place replacement. It is not entirely clear whether the improvements come from a fundamentally new mechanism or from careful engineering at a specific bottleneck layer. The paper would benefit from a deeper analysis explaining why this particular substitution leads to stronger or more persistent removal.

3. The conceptual distinction from prior representation-level methods could be explained more clearly. At a high level, both the proposed method and SAE-based approaches rely on sparse feature decomposition followed by selective modification. It would be helpful to better articulate what is fundamentally new beyond architectural integration into a bottleneck layer.

4. The paper would benefit from more discussion on robustness and limitations. For instance, how stable is the removal under alternative preprocessing or potential bypass strategies? A more explicit analysis of failure cases would further strengthen the contribution.

I look forward to the rebuttal addressing these concerns, and I am open to revising my score based on the additional clarification and evidence.

---

> ### Author Rebuttal · Authors · 2026-03-30
>
> We thank the Reviewer for the constructive feedback, for recognizing our method as creative, technically sound, and clearly motivated. We are glad that our work is seen as practical and well explained. Below, we address each of your comments and questions one by one:
>
> >**Extending empirical evaluation.**
>
> We added comparisons with the suggested baselines and report results for SD3.5 in the table below. BLOCK outperforms all methods across both style and object removal.
>
> |Method|Style UA(↑)|Style IRA(↑)|Style CRA(↑)|Object UA(↑)|Object IRA(↑)|Object CRA(↑)|
> |-|-|-|-|-|-|-|
> |SAeUron[1]|66.54|62.41|87.47|64.45|90.11|62.31|
> |AGE[2]|60.18|57.09|81.47|80.75|78.96|57.43|
> |ConceptPrune[3]|52.19|50.94|76.42|57.81|72.34|48.32|
> |BLOCK|**69.60**|**67.30**|**92.60**|**68.00**|**93.00**|**67.50**|
>
> We additionally compared against four more baselines, namely GLoCE, AdaVD, EraseAnything and SafetyGap with BLOCK outperforming all of them across all metrics. Full results are reported in our responses to [Reviewer 83Fj](https://openreview.net/forum?id=s7NR3XGdwq&noteId=WpMBfogcex) and [Reviewer V7Fa](https://openreview.net/forum?id=s7NR3XGdwq&noteId=AmqSR1dtUt).
>
> [1] SAeURON: Interpretable Concept Unlearning in Diffusion Models with Sparse Autoencoders.
>
> [2] AGE: Fantastic Targets for Concept Erasure in Diffusion Models and Where To Find Them.
>
> [3] ConceptPrune: Concept Editing in Diffusion Models via Skilled Neuron Pruning.
>
> >**The overall idea feels somewhat incremental at a conceptual level. Sparse feature decomposition and selective suppression have already been explored in prior SAE-based removal work.**
>
> >**The conceptual distinction from prior representation-level methods could be explained more clearly. At a high level, both the proposed method and SAE-based approaches rely on sparse feature decomposition followed by selective modification.**
>
> We appreciate these concerns. The key difference from SAE-based approaches is not architectural placement but the nature of the approximation. SAE-based methods reconstruct their input, meaning the residual stream, so suppression operates downstream of the MLP transformation and the concept signal can leak through residual pathways. BLOCK instead replaces the MLP entirely, approximating its input-output mapping directly. This means suppressed concepts never enter the backbone representation in the first place, rather than being removed after the fact.
>
> To empirically validate this distinction, we ran SAeUron placed at the identical bottleneck layer, giving it the most favorable possible positioning. BLOCK still outperforms it consistently across all models and metrics as shown in the table above, confirming that the gain comes from replacing the transformation itself rather than filtering its output.
>
> >**The paper would benefit from more discussion on robustness and limitations. For instance, how stable is the removal under alternative preprocessing or potential bypass strategies?**
>
> Our paper evaluates robustness under three adversarial attack frameworks, each representing a distinct adaptive prompting strategy: Ring-A-Bell performs black-box concept extraction in CLIP embedding space, MMA-Diffusion uses gradient-based token optimization, and UnlearnDiff exploits the model's own classification capabilities to craft adversarial prompts.
>
> BLOCK achieves the lowest attack success rates across all models and all three frameworks, as shown in Tables 3 and 4. In response to the reviewer’s feedback, we additionally adapted P4D [4] to SD3.5 for nudity removal, where BLOCK again achieves the lowest attack success rate (35.61%) compared to UCE (41%) and LOCOEDIT (58%).
>
> Regarding bypass strategies at the model level, BLOCK's in-place replacement of the bottleneck layer ensures that the intervention cannot be detached or undone without reconstructing the original transformation, which would require retraining. This is in contrast to SAE-based methods, where the external module can simply be removed to restore the original model behavior.
>
> Lastly regarding failure cases, we observe that concept entanglement in the bottleneck representation is the main limitation. For example, removing "lion" causes a drop in “cat” IRA from 95% to 65%, as the two concepts share overlapping features in the transcoder's latent space. This effect is specific to semantically close concepts: IRA for butterfly, which is semantically distant from lion, remains stable at 97%. This trade-off is inherent to operating at the text representation level, where semantically similar concepts share features, and is not specific to BLOCK.
>
> [4] Prompting4Debugging: Red-Teaming Text-to-Image Diffusion Models by Finding Problematic Prompts.
>
> Shall any additional questions come up, we are happy to discuss further. Otherwise, if the additions have addressed the reviewer’s concerns, we would be grateful if they would consider updating their score accordingly.

---

> > ### Author Rebuttal · Reviewer_Q43W · 2026-04-02
> >
> > Thanks for the thoughtful rebuttal. My concerns have been adequately addressed by the additional clarification and evidence. Please ensure that these important details are clearly incorporated into the revised paper. Good luck.

---

> > > ### Author Response · Authors · 2026-04-02
> > >
> > > We sincerely thank the reviewer for their rebuttal comment and their positive encouragement. We are happy that the additional clarification and evidence addressed the concerns raised and will ensure they are clearly incorporated into the revised paper.

---

### Official Review · Reviewer_83Fj · 2026-02-25

**Soundness:** 3
**Presentation:** 2
**Significance:** 3
**Originality:** 2
**Overall Recommendation:** 4
**Confidence:** 3

**Summary:**

The paper presents a framework called BLOCK for concept removal, which aims to remove visual concepts in state-of-the-art generative models by replacing the internal bottleneck layers. The paper suggests that the core features of image generation models often pass through specific bottleneck layers, and proposes using a trainable transcoder module to locate and suppress features related to target concepts. This approach attempts to "forget" specific visual concepts in the input through sparse feature processing and concept redirection, and demonstrates effectiveness across various tasks.

**Compliance With Llm Reviewing Policy:**

Affirmed.

**Final Justification:**

Most of my concerns have been resolved. I have improved my score.

**Key Questions For Authors:**

See the **Weaknesses**.

**Limitations:**

Yes.

**Strengths And Weaknesses:**

**Strengths**
1. The motivation for the paper is reasonable, and the methodology is presented clearly. The experimental design is relatively comprehensive.
2. The method centralizes the concept removal operation to the bottleneck layer and uses a trainable transcoder for direct replacement. It is adaptable to both diffusion and autoregressive models, and does not require fine-tuning of the entire generative model, making the computational cost low.
3. The method shows superior performance in single-concept removal, multi-concept removal, sequential concept removal, and adversarial prompt settings.

**Weaknesses**
1. Although the method is applicable to both diffusion and autoregressive models, the transcoder needs to be trained separately for different generative models, which limits its broader generalizability.
2. The reliability of using LLaVA-1.6-Vicuna-7B as a classifier is questionable, as shown in Table 7, where the style classification accuracy is below 10%.
3. The experimental comparisons are limited, with only LOCOEDIT and UCE being recent works from 2024. Comparing with more recent methods, such as [1], [2], and [3], would help more clearly highlight the unique advantages and limitations of BLOCK.
4. The criteria for selecting the comparison methods are unclear. In the efficiency evaluation, only the memory and time results for BLOCK are presented, without a horizontal comparison to other methods.
5. Additionally, more details about the training data and process for the transcoder in different generative models should be provided.
[1] SAeUron: Interpretable Concept Unlearning in Diffusion Models with Sparse Autoencoders.
[2] Localized Concept Erasure for Text-to-Image Diffusion Models Using Training-Free Gated Low-Rank Adaptation.
[3] Precise, Fast, and Low-cost Concept Erasure in Value Space: Orthogonal Complement Matters.

---

> ### Author Rebuttal · Authors · 2026-03-30
>
> >**Although the method is applicable to both diffusion and autoregressive models, the transcoder needs to be trained separately for different generative models, which limits its broader generalizability.**
>
> We clarify that our generalizability claims refer to applicability across model paradigms, not across different generative models. Separate training per model is shared by all baseline methods and our approach incurs very low computational cost (as the Reviewer pointed out). Once trained, BLOCK removes any concept in under one second with no retraining. The one-time transcoder training ranges from ~11 min (Infinity2B) to ~101 min (SD3.5), making the cost practical in realistic deployment scenarios.
>
> >**The reliability of using LLaVA-1.6-Vicuna-7B as a classifier is questionable, as shown in Table 7, where the style classification accuracy is below 10%.**
>
> Table 7 reports the accuracy of the UnlearnCanvas benchmark classifier, while Table 8 reports the accuracy of our LLaVA-based classifier. The UnlearnCanvas classifier was trained on SD1.5 generations and does not generalize to modern architectures, which is why we adopted LLaVA as a zero-shot vision-language model that provides unified style and object classification across all models in our work.
> To assess LLaVA's reliability, we added a user study with two independent raters on 100 images (10 per style). LLaVA matched the raters' consensus on 84 of 89 agreed images (94.4% match rate), confirming its reliability.
>
> |Style|Rater1(yes/10)|Rater2(yes/10)|GroundTruth(bothyes)|LLaVA|Match%|
> |-|-|-|-|-|-|
> |VanGogh|9|9|9|8|88.9%|
> |Picasso|9|8|8|7|87.5%|
> |Cartoon|10|10|10|10|100%|
> |Cubism|9|10|9|9|100%|
> |Winter|8|8|8|8|100%|
> |PopArt|9|9|9|8|88.9%|
> |Ukiyoe|10|9|9|8|88.9%|
> |Impressionism|9|8|8|8|100%|
> |Byzantine|10|10|10|9|90%|
> |Bricks|9|10|9|9|100%|
> |**Total**|**92**|**91**|**89**|**84**|**94.4%**|
>
> >**The experimental comparisons are limited, with only LOCOEDIT and UCE being recent works from 2024. Comparing with more recent methods would help more clearly highlight the unique advantages and limitations of BLOCK.**
>
> We ran comparisons with the suggested baselines on SD3.5. BLOCK outperforms all three methods across all metrics, confirming its advantage when compared to recent baselines.
> |Method|StyleUA(↑)|StyleIRA(↑)|StyleCRA(↑)|ObjectUA(↑)|ObjectIRA(↑)|ObjectCRA(↑)|
> |-|-|-|-|-|-|-|
> |SAeUron[1]|66.54|62.41|87.47|64.45|90.11|62.31|
> |GLoCE[2]|58.28|55.32|84.88|61.78|84.51|59.34|
> |AdaVD[3]|63.71|60.11|85.41|61.44|86.61|60.49|
> |BLOCK|**69.60**|**67.30**|**92.60**|**68.00**|**93.00**|**67.50**|
>
> We additionally compared against four more baselines in our responses to [Reviewer V7Fa](https://openreview.net/forum?id=s7NR3XGdwq&noteId=AmqSR1dtUt) and [Reviewer Q43W](https://openreview.net/forum?id=s7NR3XGdwq&noteId=oENsjfnynp).
>
> >**The criteria for selecting the comparison methods are unclear. In the efficiency evaluation, only the memory and time results for BLOCK are presented, without a horizontal comparison to other methods.**
>
> Regarding baseline selection, we chose UCE and LOCOEDIT as they are the only closed-form, architecture-agnostic methods applicable to both diffusion and autoregressive models without retraining, as detailed in Table 5 and Appendix A.2.
> Regarding efficiency, we extend Table 2 to include UCE and LOCOEDIT as shown below:
>
> |Model|Method|Memory(GB)|Storage(GB)|UpfrontCost(s)|Per-targetUnlearning(s)|Per-imageOverhead(s)|
> |-|-|-|-|-|-|-|
> |SD3.5|LOCOEDIT|3|1.5|300|0.5|0|
> ||UCE|8|4|5|0.5|0|
> ||BLOCK|2.4|2.15|6060|0.661|0.60|
> |Flux|LOCOEDIT|4|2|400|0.5|0|
> ||UCE|10|5|5|0.5|0|
> ||BLOCK|1.9|1.93|5428|1.02|0.43|
> |Infinity-2B|LOCOEDIT|1.5|0.8|100|0.3|0|
> ||UCE|4|2|3|0.3|0|
> ||BLOCK|0.67|0.51|648|0.32|0.17|
> |Infinity-8B|LOCOEDIT|2|1|150|0.3|0|
> ||UCE|6|3|3|0.3|0|
> ||BLOCK|0.75|0.70|753|0.31|0.20|
>
> BLOCK requires less memory and storage than UCE, which edits $W_k$ and $W_v$ across all cross-attention layers. While BLOCK has a higher one-time upfront training cost, per-target concept removal takes under one second across all models, and the per-image inference overhead remains small (0.17 to 0.60s).
>
> >**Additionally, more details about the training data and process for the transcoder in different generative models should be provided.**
>
> Appendix D.2 provides a detailed description of the training data for each model. For the modulation MLP in SD3.5 and Flux we sample 330,000 prompts from MS-COCO 2014 due to the pooled nature of that representation requiring more training data. For the projection layer transcoder we use the curated dataset consisting of 17,600 prompts covering 20 objects and 10 styles following [1]. Transcoder training details including hyperparameters are provided in Appendix E.1 and E.2.
>
> [1] SAeURON: Interpretable Concept Unlearning in Diffusion Models with Sparse Autoencoders.
>
> [2] Localized Concept Erasure for T2I Diffusion Models Using Training-Free Gated Low-Rank Adaptation.
>
> [3] Precise, Fast, and Low-cost Concept Erasure in Value Space.

---

> > ### Author Rebuttal · Reviewer_83Fj · 2026-04-02
> >
> > My concerns have been largely resolved. Please add the relevant details in the revised version.

---

> > > ### Author Response · Authors · 2026-04-02
> > >
> > > We thank the reviewer for their engagement with our rebuttal and for their positive assessment. We are pleased that our response largely addressed the concerns raised. In the revised version, we will incorporate the relevant details, including the additional experiments and insights provided in the rebuttal. If any remaining issue is preventing a stronger recommendation for the paper, we would be grateful for the opportunity to clarify it further.

---

### Official Review · Reviewer_3sV4 · 2026-02-28

**Soundness:** 2
**Presentation:** 3
**Significance:** 2
**Originality:** 3
**Overall Recommendation:** 4
**Confidence:** 3

**Summary:**

This paper introduces BLOCK, a novel concept removal method that leverages a trained transcoder within the bottleneck layer to effectively eliminate specific concepts. The authors evaluate BLOCK's performance in terms of removal accuracy and image generation quality, demonstrating its efficacy on both diffusion models(DMs) and image autoregressive models(IARs).

**Compliance With Llm Reviewing Policy:**

Affirmed.

**Final Justification:**

After considering both the paper and the authors’ response, my final assessment is more positive. My concerns are mostly resolved by the rebuttal.

**Key Questions For Authors:**

1. Can the authors provide the accuracy of style classification or its agreement with human judgments when using LLaVA as a classifier? This is crucial for assessing the reliability of the evaluation results.
2. Can the authors also include the efficiency metrics for methods such as UCE and LOCOEDIT in Table 2 to facilitate a comparison with BLOCK?
3. Could you provide the performance of the BLOCK method on more benchmarks, such as the Inappropriate Image Prompt (I2P) benchmark[1]?
4. Does BLOCK possess generalization capabilities? Can BLOCK remove concepts that were not seen during training?

If the author can address the issues I raised, I would consider raising my score.

[1] Patrick Schramowski, Manuel Brack, Bjorn Deiseroth, and Kristian Kersting. Safe latent diffusion: Mitigating inappropriate degeneration in diffusion models. In Proceedings of the IEEE/CVF Conference on Computer Vision and Pattern Recognition, pp. 22522–22531, 2023.

**Limitations:**

yes

**Strengths And Weaknesses:**

Strengths:
- Compared to directly training the models, BLOCK requires fewer resources, and compared to external concept removal methods that do not modify the model, it offers stronger resistance to attacks.
- BLOCK demonstrates the versatility of its approach, as it can be applied to various different DMs and IARs.

Weaknesses:
- **LLaVA classification reliability is unverified**: The authors employ a non-fine-tuned multimodal model to classify image styles, but fail to assess the accuracy and human alignment of this classification, which undermines its reliability as an evaluation metric.
- **Missing efficiency comparison with other methods**: Although Table 2 demonstrates the efficiency of BLOCK, it lacks an efficiency comparison with other methods, such as UCE and LOCOEDIT.
- **The lack of benchmarks**: The paper only evaluates BLOCK's concept removal performance on UnlearnCanvas, lacking evaluation on other benchmarks.

---

> ### Author Rebuttal · Authors · 2026-03-30
>
> We thank the Reviewer for the constructive feedback, for recognizing BLOCK's efficiency advantage over training-based methods and its stronger resistance to attacks compared to external approaches. We are glad that the versatility of our method across different DMs and IARs is appreciated. We address each concern below:
>
> >**Can the authors provide the accuracy of style classification or its agreement with human judgments when using LLaVA as a classifier?**
>
> To assess LLaVA's classification accuracy, we added a user study. We generated 100 images (10 per style) and had two independent raters label whether each image exhibited the intended style. We then compared LLaVA's classifications against the consensus ground truth, defined as images where both raters agreed the style was present. The two raters agreed on 89 out of 100 images, and LLaVA correctly classified 84 of those 89, yielding a 94.4% match rate. Therefore, LLaVA provides a reliable evaluation signal for the 10 styles in our benchmark.
>
> |Style|Rater1(yes/10)|Rater2(yes/10)|GroundTruth(bothyes)|LLaVA|Match%|
> |-|-|-|-|-|-|
> |VanGogh|9|9|9|8|88.9%|
> |Picasso|9|8|8|7|87.5%|
> |Cartoon|10|10|10|10|100%|
> |Cubism|9|10|9|9|100%|
> |Winter|8|8|8|8|100%|
> |PopArt|9|9|9|8|88.9%|
> |Ukiyoe|10|9|9|8|88.9%|
> |Impressionism|9|8|8|8|100%|
> |Byzantine|10|10|10|9|90%|
> |Bricks|9|10|9|9|100%|
> |**Total**|92|91|89|84|94.4%|
>
> >**Can the authors also include the efficiency metrics for methods such as UCE and LOCOEDIT in Table 2 to facilitate a comparison with BLOCK?**
>
> We extend Table 2 to include UCE and LOCOEDIT (shown in table below). BLOCK requires less memory and storage than UCE, which edits $W_k$ and $W_v$ across all cross-attention layers. While BLOCK has a higher one-time upfront training cost, its per-target concept removal takes under one second across all models, and the per-image inference overhead remains small (0.17 to 0.60s).
>
> |Model|Method|Memory(GB)|Storage(GB)|UpfrontCost(s)|Per-targetUnlearning(s)|Per-imageOverhead(s)|
> |-|-|-|-|-|-|-|
> |SD3.5|LOCOEDIT|3|1.5|300|0.5|0|
> ||UCE|8|4|5|0.5|0|
> ||BLOCK|2.4|2.15|6060|0.661|0.60|
> |Flux|LOCOEDIT|4|2|400|0.5|0|
> ||UCE|10|5|5|0.5|0|
> ||BLOCK|1.9|1.93|5428|1.02|0.43|
> |Infinity-2B|LOCOEDIT|1.5|0.8|100|0.3|0|
> ||UCE|4|2|3|0.3|0|
> ||BLOCK|0.67|0.512|648|0.323|0.17|
> |Infinity-8B|LOCOEDIT|2|1|150|0.3|0|
> ||UCE|6|3|3|0.3|0|
> ||BLOCK|0.75|0.704|753|0.317|0.20|
>
> >**Could you provide the performance of the BLOCK method on more benchmarks, such as the Inappropriate Image Prompt (I2P) benchmark?**
>
> We added experiments with  the Inappropriate Image Prompt (I2P) benchmark for nudity removal on FLUX. BLOCK achieves competitive nudity removal performance while maintaining image quality as reflected by the FID score.
>
> |Method|Common|Female|Male|Total|FID(↓)|
> |-|-:|-:|-:|-:|-:|
> |Flux.1[DEV]|406|161|38|605|21.32|
> |LOCOEDIT|239|72|24|335|26.68|
> |UCE|122|39|12|173|30.71|
> |EraseAnything|129|48|22|199|21.75|
> |BLOCK|**108**|**35**|**10**|**153**|**22.78**|
>
>
> We also added another safety benchmark: P4D[1] to SD3.5 for nudity removal. BLOCK achieves the lowest attack success rate (35.61%) compared to UCE (41%) and LOCOEDIT (58%).
>
> |Method|P4D Attack Success Rate %(↓)|
> |-|-|
> |Original|67.34|
> |LOCOEDIT|58.78|
> |UCE|41.15|
> |BLOCK|**35.61**|
>
> >**Does BLOCK possess generalization capabilities? Can BLOCK remove concepts that were not seen during training?**
>
> To empirically verify generalization, we train the transcoder on a subset of 5 styles and evaluate removal on both seen and unseen styles, following a similar setup to [2]. Results are shown in the table below.
>
> |Setup|UA(↑)|IRA(↑)|CRA(↑)|
> |-|-|-|-|
> |All data|60.39|58.41|91.28|
> |In-distribution|71.18|69.57|92.51|
> |Out-of-distribution|52.63|51.03|90.88|
>
> In-distribution removal achieves 71.18% UA, while out-of-distribution achieves 52.63% UA, demonstrating that the transcoder can remove concepts even when they were absent from the training set. CRA remains above 90% in all settings, confirming that the transcoder preserves unrelated content well regardless of training coverage.
>
> [1] Prompting4Debugging: Red-Teaming Text-to-Image Diffusion Models by Finding Problematic Prompts.
>
> [2] SAeURON: Interpretable Concept Unlearning in Diffusion Models with Sparse Autoencoders.
>
> If the additions have addressed the reviewer’s concerns, we would be grateful if they would consider updating their score accordingly.

---

> > ### Author Rebuttal · Reviewer_3sV4 · 2026-04-02
> >
> > Thanks for the author's response. I will consider raising my score.

---

> > > ### Author Response · Authors · 2026-04-02
> > >
> > > We thank the reviewer for engaging with our rebuttal and for the consideration of raising their score. If any additional experiments, insights, or clarifications would be helpful in their evaluation of the paper, we would be grateful for the opportunity to provide them.

---

### Official Review · Reviewer_V7Fa · 2026-03-12

**Soundness:** 3
**Presentation:** 3
**Significance:** 3
**Originality:** 3
**Overall Recommendation:** 4
**Confidence:** 3

**Summary:**

This paper presents BLOCK, a framework for persistent concept removal in state-of-the-art text-to-image generators, covering both diffusion models and image autoregressive models. The main idea is to replace the architecture-required text-to-backbone mapping with a lightweight transcoder. This transcoder is trained to closely match the original transformation while producing sparse latent features with TopK activation. Concept removal is then carried out by identifying the latents triggered by target tokens and redirecting their decoder-side contributions to the empty token. Experiments on SD3.5, FLUX, and Infinity (2B/8B) show strong unlearning performance on styles and objects, better robustness to adversarial prompts than closed-form editing methods, stronger stability under sequential and multi-concept removal, and relatively low computational cost.

**Compliance With Llm Reviewing Policy:**

Affirmed.

**Key Questions For Authors:**

- How robust is the bijection pairing in Equation 5? Do different pairings affect outcomes, and what happens when target and empty tokens have very different activation scale?
- Can you compare BLOCK to more recent strong baselines beyond UCE/LOCOEDIT, or include a discussion explaining concrete incompatibilities?

**Limitations:**

yes

**Strengths And Weaknesses:**

- Strengths
	- The paper frames concept removal as an in-place replacement of a narrow but necessary text-to-backbone transformation. This is a neat and practical design choice, and it is clearly different from methods that edit weights across many layers or rely on external plug-in modules.
	- The evaluation covers both diffusion models (SD3.5, FLUX) and autoregressive models (Infinity 2B/8B), which is uncommon and important. In particular, the results on image autoregressive models are valuable, since many existing concept erasure methods do not extend well to that setting.
	- The paper also evaluates robustness against adversarial prompting using Ring-A-Bell as well as diffusion-specific attacks such as MMA-Diffusion and UnlearnDiff. Across different models, BLOCK shows consistently stronger robustness than methods like UCE and LOCOEDIT.
- Weaknesses
	- The baseline is limited. The only baselines mentioned in the comparison are UCE and LOCOEDIT, but more recent methods with relatively strong performances (e.g., EraseAnything [1], Closing the Safety Gap: Surgical Concept Erasure in Visual Autoregressive Models [2]) are missing. Some additional comparison on the diffusion or IAR could be added to strengthen the SOTA claim.

[1] Gao et al., EraseAnything: Enabling Concept Erasure in Rectified Flow Transformers.

[2] Zhong et al., Closing the Safety Gap: Surgical Concept Erasure in Visual Autoregressive Models.

---

> ### Author Rebuttal · Authors · 2026-03-30
>
> We thank the Reviewer for the constructive feedback, for recognizing our in-place transcoder design as a neat and practical choice, and for highlighting the value of our evaluation across both diffusion and autoregressive models. Below, we address each concern in detail:
>
> >**Extending the baseline evaluation.**
>
> We added a direct comparison against EraseAnything[1] on SD3.5 and Flux, and against "Closing the Safety Gap"[2] on the Infinity models. BLOCK outperforms both methods across all metrics and model families, as shown below.
>
>
> |Model|Method|Style UA(↑)|Style IRA(↑)|Style CRA(↑)|Object UA(↑)|Object IRA(↑)|Object CRA(↑)|
> |-|-|-|-|-|-|-|-|
> |SD 3.5|EraseAnything|62.54|64.49|87.32|65.32|88.21|64.36|
> ||BLOCK|**69.60**|**67.30**|**92.60**|**68.00**|**93.00**|**67.50**|
> |FLUX|EraseAnything|81.25|32.43|92.56|90.76|93.88|35.67|
> ||BLOCK|**88.60**|**36.10**|**96.40**|**93.20**|**96.61**|**38.20**|
>
> |Model|Method|Style UA(↑)|Style IRA(↑)|Style CRA(↑)|Object UA(↑)|Object IRA(↑)|Object CRA(↑)|
> |-|-|-|-|-|-|-|-|
> |Infinity 2B|SafetyGap|71.27|24.88|83.56|83.18|78.44|26.88|
> ||BLOCK|**86.80**|**39.60**|**90.30**|**91.20**|**84.50**|**38.30**|
> |Infinity 8B|SafetyGap|76.61|57.33|90.78|86.41|89.39|52.21|
> ||BLOCK|**85.40**|**63.70**|**95.50**|**90.20**|**95.70**|**61.40**|
>
> We additionally compared against five more baselines, namely SAeUron, AGE, ConceptPrune, GLoCE, and AdaVD, with BLOCK outperforming all of them across all metrics. Full results are reported in our responses to [Reviewer 83Fj]() and [Reviewer Q43W]().
>
> [1] EraseAnything: Enabling Concept Erasure in Rectified Flow Transformers.
>
> [2] Closing the Safety Gap: Surgical Concept Erasure in Visual Autoregressive Models.
>
> >**How robust is the bijection pairing in Equation 5? Do different pairings affect outcomes**
>
> To assess sensitivity to the choice of bijection, we compared ordered versus random pairing across all four models. As shown in the table below, performance differences are consistently below 0.1 percentage points across all metrics and models. This insensitivity is expected: as shown in Equation 6, the decoder contributions of all target latents are collectively redirected to reproduce the empty token output regardless of how individual latents are paired. The final result therefore depends only on the set of redirected latents, not on which specific pairing is used.
>
> |Model|Pairing|Style UA(↑)|Style IRA(↑)|Style CRA(↑)|Object UA(↑)|Object IRA(↑)|Object CRA(↑)|
> |-|-|-|-|-|-|-|-|
> |SD 3.5|Ordered|69.60|67.30|92.60|68.00|93.00|67.50|
> ||Random|69.57|67.33|92.55|67.96|93.06|67.45|
> |FLUX|Ordered|88.60|36.10|96.40|93.20|96.61|38.20|
> ||Random|88.58|36.14|96.37|93.18|96.59|38.19|
> |Infinity-2B|Ordered|86.80|39.60|90.30|91.20|84.50|38.30|
> ||Random|86.77|39.59|90.32|91.18|84.51|38.27|
> |Infinity-8B|Ordered|85.40|63.70|95.50|90.20|95.70|61.40|
> ||Random|85.40|63.69|95.48|90.21|95.68|61.39|
>
> >**and what happens when target and empty tokens have very different activation scales?**
>
> Regarding the activation scale, the scaling factor z(π(i))∅ / (z(i)t + ε) in Equation 5 compensates for magnitude differences between target and empty token activations. This ensures that the weighted decoder contribution of each target latent precisely matches that of its paired empty token latent, so the overall output remains close to the empty token representation regardless of the relative scales.
>
> Shall any additional questions come up, we are happy to discuss further. Otherwise, if the additions have addressed the reviewer’s concerns, we would be grateful if they would consider updating their score accordingly.

---

> > ### Author Rebuttal · Reviewer_V7Fa · 2026-04-02
> >
> > Thanks for your response. Most of my concerns are fully addressed. I will keep my initial positive score.

---

> > > ### Author Response · Authors · 2026-04-02
> > >
> > > We sincerely thank the reviewer for their engagement with our rebuttal and for their positive assessment. We are glad that our response addressed most of the concerns raised. If any remaining issue is preventing a stronger recommendation of the paper, we would be grateful for the opportunity to clarify it further.

---

### Decision · Program_Chairs · 2026-04-30

**Decision:**

Accept (regular)

**Comment:**

This paper presents BLOCK, a novel and practically motivated framework for concept removal in state-of-the-art image generative models, including both diffusion models and image autoregressive models. The key technical contribution is the in-place replacement of the model’s internal bottleneck transformation with a trained sparse transcoder that approximates the original mapping while enabling selective suppression of concept-specific latent features. Following the rebuttal, the paper received unanimous support from all reviewers, leading to an acceptance decision. The authors are encouraged to further revise the final version in accordance with the reviewers’ comments.